# Visual Prompting Upgrades Neural Network Sparsification: A Data-Model Perspective

## Abstract

The rapid development of large-scale deep learning models questions the affordability of hardware platforms, which necessitates the pruning to reduce their computational and memory footprints. Sparse neural networks as the product, have demonstrated numerous favorable benefits like low complexity, undamaged generalization, *etc*. Most of the prominent pruning strategies are invented from a *model-centric* perspective, focusing on searching and preserving crucial weights by analyzing network topologies. However, the role of data and its interplay with model-centric pruning has remained relatively unexplored. In this research, we introduce a novel *data-model co-design* perspective: to promote superior weight sparsity by learning important model topology and adequate input data in a synergetic manner. Specifically, customized **V**isual **P**rompts are mounted to upgrade neural **N**etwork **s**parsification in our proposed **VPNs** framework. As a pioneering effort, this paper conducts systematic investigations about the impact of different visual prompts on model pruning and suggests an effective joint optimization approach. Extensive experiments with 3 network architectures and 8 datasets evidence the substantial performance improvements from **VPNs** over existing start-of-the-art pruning algorithms. Furthermore, we find that subnetworks discovered by **VPNs** from pre-trained models enjoy better transferability across diverse downstream scenarios. These insights shed light on new promising possibilities of data-model co-designs for vision model sparsification. Codes are in the supplement.

## 1 Introduction

Large-scale neural networks like vision and language models (Brown et al., 2020; Radford et al., 2019; Touvron et al., 2023; Chiang et al., 2023; Li et al., 2022; Bai et al., 2023) have attracted stupendous attention in nowadays deep learning community, which pose significantly increased demands to computing resources. While remarkable performance has been offered, they suffer from prohibitively high training and inference costs, and the deployment of these gigantic models entails substantial memory and computational overhead. For instance, inferring the GPT-3 with 175B parameters requires at least five 80GB A100 GPUs (Frantar & Alistarh, 2023).

To establish economic and lightweight network alternatives, model compression serves as an effective tool, gaining great popularity (Dettmers et al., 2022; Frantar & Alistarh, 2023; Yao et al., 2022; Sun et al., 2023; Jaiswal et al., 2023). Among plenty of efforts for compression, model pruning (LeCun et al., 1989; Gale et al., 2019; Frankle & Carbin, 2018; Chen et al., 2020) is one of the dominant approaches, aiming to trim down the least significant weights without hurting model performance. It is usually applied subsequent to the convergence of training (Frankle & Carbin, 2018; Chen et al., 2020; Molchanov et al., 2016; Wang et al., 2023; Mo et al., 2023), during the training process (Zhu & Gupta, 2017; Gale et al., 2019; Chen et al., 2021a), and even prior to the initiation of training (Mocanu et al., 2018; Lee et al., 2019; Evci et al., 2020; Wang et al., 2020b). The resulting sparsity ranges from fine-grained elements like individual weights (Zhu & Gupta, 2017) to coarse-grained structures such as neurons (Hu et al., 2016), blocks (Lagunas et al., 2021), filters (Yin et al., 2023), and attention heads (Shim et al., 2021). It is worth mentioning that the majority, if not all, of the conventional pruning algorithms, produce sparse neural networks in a *model-centric* fashion – analyzing architecture topologies and capturing their key subset by learning parameterized weight masks (Sehwag et al., 2020) or calculating proxy heuristics based on training dynamics (Han et al., 2015), architecture properties (Hoang et al., 2023), *etc*.

Thanks to the recent advances in large language models (LLMs), the *data-centric* AI regains a spotlight. Techniques like in-context learning (Brown et al., 2020; Shin et al., 2020; Liu et al., 2022a) and prompting (Liu et al., 2023a; Li & Liang, 2021) construct well-designed prompts or input templates to empower LLMs and reach impressive performances on a variety of tasks. It evidences that such data-centric designs effectively extract and compose knowledge in learned models (Wei et al., 2022), which might be a great assistant to locating critical model topologies. Nevertheless, the influence of data-centric methods on network sparsification has been less studied. To our best knowledge, only one concurrent work (Xu et al., 2023) has explored the possibility of learning *post-pruning prompts* to recover compressed LLMs. Thus, We focus on a different aim:

*How to leverage prompts to upgrade vision model sparsification, from a data-model perspective?*

Note that the effect of visual prompts on sparse vision models remains mysterious. Also, visual prompts are inherently more complex to comprehend and typically pose greater challenges in terms of both design and learning, in comparison to their linguistic counterparts.

To answer the above research questions, we start with a systematical pilot investigation of existing post-pruning prompts (Xu et al., 2023) on sparse vision models. As presented in Section 2.1, **directly inserting *post-pruning* visual prompts into sparse vision models does not necessarily bring performance gains.** To unlock the capacity of visual prompts in sparse vision models, we propose a *data-model co-design* paradigm, which jointly optimizes inputs and sparse models in the sparsification process. Specifically, we propose **VPNs** (**V**isual **P**rompting Upgrades **N**etworks **S**parsification) that co-trains the visual prompts with parameterized weight masks, exploring superior subnetworks. In a nutshell, our innovative efforts are unfolded along with the following five thrusts:

⋆ (A Pilot Study) We conduct a pilot study of post-pruning prompts in sparse vision models and surprisingly find its inefficacy in improving the performance of well fine-tuned sparse vision models.

⋆ (Algorithm) To unlock the potentials of visual prompts in vision model sparsification, we propose a novel *data-model co-design* sparsification paradigm, termed **VPNs**, which simultaneously optimizes weight masks and tailored visual prompts. It crafts appropriate visual prompts for mining improved sparse vision models.

⋆ (Experiments) We conduct extensive experiments across diverse datasets, architectures, and pruning regimes. Empirical results consistently highlight the impressive advancement of both performance and efficiency brought by **VPNs**. For example, **VPNs** outperforms the previous state-of-the-art (SoTA) methods {HYDRA (Sehwag et al., 2020), BiP (Zhang et al., 2022a), LTH (Chen et al., 2021b)} by {$3.41\%, 1.69\%, 2.00\%$} at 90% sparsity with ResNet-18 on Tiny-ImageNet.

⋆ (Extra Findings) More interestingly, we demonstrate that the sparse masks from our **VPNs** enjoy enhanced transferability across multiple downstream tasks.

⋆ (Potential Practical Benefits) **VPNs** can be seamlessly integrated into structured pruning approaches, enabling more real-time speedups and memory reduction with competitive accuracies.

## 2 RELATED WORKS AND A PILOT STUDY

**Neural Network Pruning.** Pruning (Mozer & Smolensky, 1989; LeCun et al., 1989) aims at compressing networks by removing the least important parameters in order to benefit the model generalization (Chen et al., 2022c), robustness (Sehwag et al., 2020), stability (Hooker et al., 2020), transferability (Chen et al., 2020), *et al*. In the literature, an unpruned network is often termed the "dense network", while its compressed counterpart is referred to as a "subnetwork" of the dense network (Chen et al., 2021b). A commonly adopted compression strategy follows a three-phase pipeline: pre-training, pruning, and fine-tuning. Categorizing based on this pipeline, pruning algorithms can be segmented into post-training pruning, during-training pruning, and prior-training pruning. *Post-training* pruning methods, applied after the dense network converges, are extensively explored. In general, these methods fall under three primary umbrellas: weight magnitude-based techniques (Han et al., 2015), gradient-centric methods (Molchanov et al., 2016; Sanh et al., 2020; Jiang et al., 2021), and approaches leveraging Hessians (LeCun et al., 1989; Hassibi & Stork, 1992; Dong et al., 2017). Along with the rising of foundational models, more innovative post-training pruning methods have emerged to amplify their resource-efficiency (Zafrir et al., 2021; Peng et al., 2022b; Lagunas et al., 2021; Frantar et al., 2021; Peng et al., 2021; 2022a; Chen et al., 2022a). *During-training* pruning, which is introduced by (Finnoff et al., 1993), presents an effective variant

for model sparsification. It begins by training a dense model and then iteratively trims it based on pre-defined criteria, until obtaining the desired sparse subnetwork. Significant contributions to this approach category are evident in works such as (Zhu & Gupta, 2017; Gale et al., 2019; Chen et al., 2022b; Huang et al., 2022). As a more intriguing yet challenging alternative, *prior-training* pruning thrives (Huang et al., 2023; Jaiswal et al., 2023), which targets to identify the optimal subnetwork before fine-tuning the dense network. Mocanu et al. (2018); Dettmers & Zettlemoyer (2019); Evci et al. (2020); Schwarz et al. (2021) take a step further to advocate one particular group of sparse neural networks that are extracted at random initialized models, trained from the ground up, and able to reach commendable results.

**Prompting.** Traditionally, the quest for peak performance is centered on manipulating model weights. However, prompting heralds a pivotal shift towards *data-centric* studies, illuminating the potential of innovative input design. The concept of prompting emerges in the domain of natural language processing (NLP) as a proficient approach for adapting pre-trained models to downstream tasks (Liu et al., 2023a). Specifically, GPT-3 showcases its robustness and generalization to downstream transfer learning tasks when equipped with handpicked text prompts, especially in settings like few-shot or zero-shot learning (Brown et al., 2020). There is a significant amount of work around refining text prompting, both in terms of crafting superior prompts (Shin et al., 2020; Jiang et al., 2020) and representing prompts as task-specific continuous vectors (Liu et al., 2021). The latter involves optimizing these prompts using gradients during the fine-tuning phase, which is termed Prompt Tuning (Li & Liang, 2021; Lester et al., 2021; Liu et al., 2021). Interestingly, this approach rivals the performance of full fine-tuning but enjoys the advantage of high parameter efficiency and low storage cost. The design philosophy of prompt tuning is extended to the computer vision realm by Bahng et al. (2022), which incorporates prompt parameters directly into input images, thereby crafting a prompted image, termed Visual Prompt (VP). Building on this foundation, Jia et al. (2022) proposes a visual-prompt tuning method that modifies pre-trained Vision Transformer models by integrating selected parameters into the Transformer's input space. Chen et al. (2023) reveals the importance of correct label mapping between the source and the target classes and introduces iterative label mapping to help boost the performance of VP. Further advancements are made by Liu et al. (2023c); Zheng et al. (2022); Zhang et al. (2022b), which devise a prompt adapter towards enhancing or pinpointing an optimal prompt for a given domain. In a parallel approach, Zang et al. (2022); Zhou et al. (2022b) and Zhou et al. (2022a) introduce visual prompts in conjunction with text prompts to vision-language models, resulting in a noted improvement in downstream performance.

## 2.1 A PILOT STUDY

**Motivation.** The question of whether pruning should be either a more *model-centric* or *data-centric* process continues to be debated within the field. Certain proponents suggest pruning as *model-centric*, with their assertions bolstered by the success of approaches like Syn-Flow (Tanaka et al., 2020) which, despite not using any real data pass, deliver performances akin to dense networks. Yet, a considerable body of research contradicts this, emphasizing the superiority of post-training pruning techniques over prior-training ones, thereby

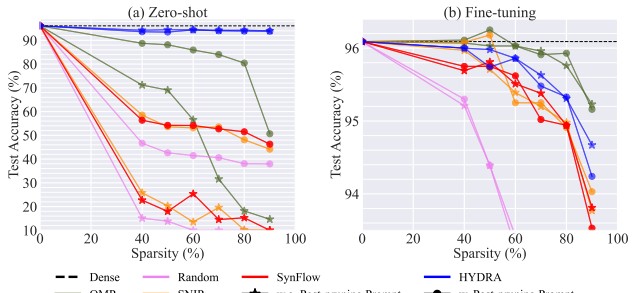

Figure 1: **Post-pruning Prompt Results.** Performance of 5 pruning methods and their post-pruning prompt counterparts on ResNet-18 and CIFAR10, which are marked as ● and ★, respectively. The dashed line indicates the dense network's performance. (a) Post-pruning with zero-shot. (b) Post-pruning with fine-tuning. *Post-pruning prompt is only valid without fine-tuning.*

articulating pruning's dependence on data (Zhang et al., 2022a; Liu et al., 2023b). To further complicate matters, the rise of LLMs has underscored the central role of data in shaping NLP's evolution. New strategies like in-context learning and prompting, designed to enhance LLMs' task-specific performance, have come to the fore. However, the precise role of *data-centric* designs in sparsification remains an under-explored area, meriting further attention.

To the best of our knowledge, Xu et al. (2023) is the sole concurrent study to delve into the potential of harnessing prompts to recover compressed LLMs. This research illuminates the efficacy of

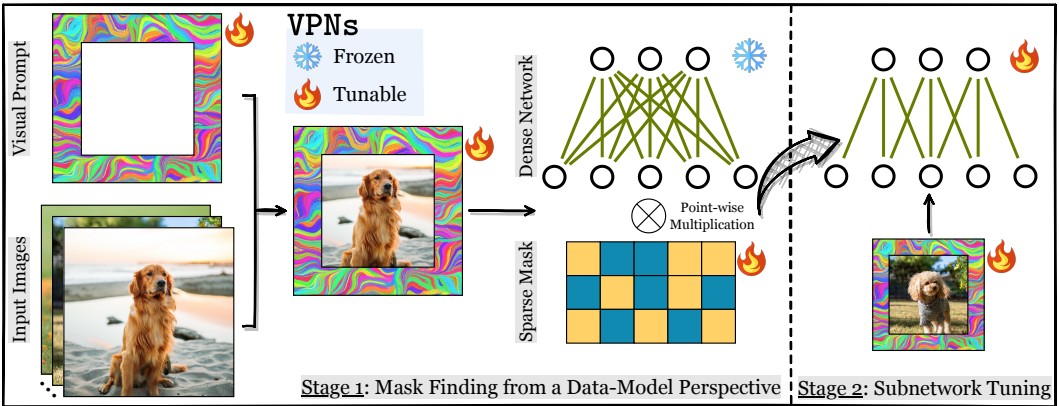

Figure 2: Overview of **VPNs**. In stage 1, it locates superior sparse topologies from a data-model perspective. A tailored VP is added to input samples and weight masks are jointly optimized together with the VP. In stage 2, the identified subnetwork is further fine-tuned with its VP.

*post-pruning* prompts, both manually crafted and learned "soft" prompts, in enhancing the performance of compressed LLMs. However, the influence of VP on vision model sparsification presents an enigma, as VP is inherently more intricate and poses distinct challenges in designing and learning relative to their textual counterparts (Bahng et al., 2022). To demystify it, we first investigate the post-pruning prompts on sparsified vision models. The experiments are conducted on ImageNet-1K pre-trained ResNet-18 (He et al., 2016) and CIFAR100 (Krizhevsky et al., 2009). We adopt 5 pruning methods, *i.e.*, Random (Liu et al., 2022b), OMP (Han et al., 2015), SNIP (Lee et al., 2019), SynFlow (Tanaka et al., 2020), and HYDRA (Sehwag et al., 2020), to analyze the performance of post-pruning prompts across various sparsity levels. To make a holistic study, we apply the post-pruning prompt to the sparse models with and without fine-tuning the subnetwork, referred to as **"Zero-shot"** and **"Fine-tuning"**, respectively. As shown in Figure 1, we find that: Post-pruning prompts only escalate the subnetworks before fine-tuning and bring marginal gains to the subnetwork with fine-tuning. The reason is likely that, after fine-tuning, the sparse model is sufficiently strong, leaving less room for prompts to enhance its performance. Neither of these settings consistently surpasses the standard no-prompting approach, which involves pruning and fine-tuning.

**Open Question.** As deliberated, the post-pruning prompting paradigm falls short in improving sparse vision models. This situation compels us to ask – *how to effectively utilize visual prompts to enhance the sparsification of vision models?* Our answer: a **data-model co-design** paradigm.

## 3 METHODOLOGY

In this section, we provide details about **VPNs**, which contains (1) designing appropriate visual prompts and (2) incorporating VPs to upgrade the sparse training of vision models in a **data-model** jointly optimization manner. An overview of our proposed **VPNs** is depicted in Figure 2.

### 3.1 DESIGNING APPROPRIATE VISUAL PROMPTS

Visual prompts are proposed to address the problem of adapting a pre-trained source model to downstream tasks without any task-specific model modification, *e.g.* fine-tuning network weights. To be specific, VP modifies the input image by injecting a small number of learnable parameters. Let $\mathcal{D} = \{(\mathbf{x}_1, y_1), ..., (\mathbf{x}_n, y_n)\}$ denotes the vanilla downstream image dataset, $\mathbf{x}$ is an original image in $\mathcal{D}$ with $y$ as its label, and $n$ represents the total number of images. The generic form of input prompting is then formulated as:

$$\mathbf{x}'(\boldsymbol{\delta}) = h(\mathbf{x}, \boldsymbol{\delta}), \mathbf{x} \in \mathcal{D} = \{(\mathbf{x}_1, y_1), ..., (\mathbf{x}_n, y_n)\}, \tag{1}$$

where $h(\cdot, \cdot)$ is an input transformation that integrates $\mathbf{x}$ with the learnable input perturbation $\boldsymbol{\delta}$ and $\mathbf{x}'$ is the modified data after prompting.

Our VP design first resizes the original image $\mathbf{x}$ to a specific **input size** $i \times i$ and pad it to $224 \times 224$ with $0$ values to get the resized image. We mark this process as $r^i(\mathbf{x})$, where $r(\cdot)$ refers to the resize and pad operation and $i$ indicates the target size, *i.e.* input size. Subsequently, we initiate the perturbation parameters of $\boldsymbol{\delta}$ as a $224 \times 224$ matrix and mask a portion of them. Different visual prompts can be crafted by

Figure 3: Our VP.

masking parameters in diverse shapes, locations, and sizes. In our case, the fixed mask is a central square matrix and the left four peripheral segments stay tunable. This kind of perturbation design is similar to *pad prompt* in Bahng et al. (2022) and the width of each peripheral side marked as $p$ is called **pad size**. More details about the prompt can be found in Appendix B. Finally, the input prompting operation of **VPNs** is described as below (Figure 3):

$$\mathbf{x}'(\boldsymbol{\delta}) = h(\mathbf{x}, \boldsymbol{\delta}) = r^i(\mathbf{x}) + \boldsymbol{\delta}^p, \mathbf{x} \in \mathcal{D}, \tag{2}$$

where $\boldsymbol{\delta}^p$ is the pad prompt perturbation with a pad size of $p$. Note that, usually, $i + 2p$ is larger than the input sample size like 224 to sufficiently utilize all sample pixels.

## 3.2 Upgrading Network Sparsification with Visual Prompts

Given the input prompt formulation (Equation 2), VP seeks to advance downstream task performance of a pre-trained source model $f_{\boldsymbol{\theta}_{\text{pre}}}$ by optimizing the tunable part in $\boldsymbol{\delta}$. Here $\boldsymbol{\theta}_{\text{pre}}$ refers to the pre-trained weights that are fixed in this stage. It raises a *prompt optimization problem* as follows:

$$\underset{\boldsymbol{\delta}}{\text{minimize}} \quad \mathbb{E}_{(\mathbf{x},y)\in\mathcal{D}}\mathcal{L}(f_{\boldsymbol{\theta}_{\text{pre}}}(\mathbf{x}'(\boldsymbol{\delta})), y), \tag{3}$$

where $\mathcal{L}$ is the objective function such as a cross-entropy loss for image recognition problems. As for the network sparsification, we recast it as an empirical risk minimization with respect to a learnable parameterized mask and the corresponding model weights can be frozen. Then a *mask finding problem* is depicted as below:

$$\underset{\mathbf{m}}{\text{minimize}} \quad \mathbb{E}_{(\mathbf{x},y)\in\mathcal{D}}\mathcal{L}(f_{\boldsymbol{\theta}_{\text{pre}}\odot\mathbf{m}}(\mathbf{x}), y), \quad s.t. \ ||\mathbf{m}||_0 \leq (1-s)|\boldsymbol{\theta}_{\text{pre}}|, \tag{4}$$

where $\mathbf{m}$ is the mask variable, $\boldsymbol{\theta}_{\text{pre}} \odot \mathbf{m}$ is a point-wise multiplication between the mask and model weights, $s$ denotes the desired sparsity level, and $|\boldsymbol{\theta}_{\text{pre}}|$ refers to the number of parameters in $\boldsymbol{\theta}_{\text{pre}}$.

Our proposed **VPNs** leverages visual prompts to upgrade the process of model sparsification by seamlessly integrating Equations 3 and 4. To be specific, the joint optimization of prompt $\boldsymbol{\delta}$ and $\mathbf{m}$ is described as follows:

$$\underset{\mathbf{m},\boldsymbol{\delta}}{\text{minimize}} \quad \mathbb{E}_{(\mathbf{x},y)\in\mathcal{D}}\mathcal{L}(f_{\boldsymbol{\theta}_{\text{pre}}\odot\mathbf{m}}(\mathbf{x}'(\boldsymbol{\delta})), y) \quad s.t. \ ||\mathbf{m}||_0 \leq (1-s)|\boldsymbol{\theta}_{\text{pre}}|, \tag{5}$$

where the learned mask $\mathbf{m}$ will be turned into a binary matrix in the end. The thresholding technique from Ramanujan et al. (2020) is applied to map large and small scores to 1 and 0, respectively.

After obtaining sparse subnetworks from **VPNs**, a subsequent retraining phase is attached. It is another data-model co-optimization problem of VP and model weights as below:

$$\underset{\boldsymbol{\delta},\boldsymbol{\theta}}{\text{minimize}} \quad \mathbb{E}_{(\mathbf{x},y)\in\mathcal{D}}\mathcal{L}(f_{\boldsymbol{\theta}\odot\mathbf{m}}(\mathbf{x}'(\boldsymbol{\delta})), y) \quad s.t. \ \mathbf{m} = \mathbf{m}_s, \tag{6}$$

where $\boldsymbol{\theta}$ is the model parameters that are initialized as $\boldsymbol{\theta}_{\text{pre}}$. $\mathbf{m}_s$ represents the mask found by Equation 5, and is fixed in this stage.

## 3.3 Overall Procedure of VPNs

Our **VPNs** first creates a VP following the Equation 2. Then, to locate the **VPNs** sparse subnetwork, VP and the parameterized mask are jointly optimized based on Equation 5. In this step, $\mathbf{m}$ is initialized with a scaled-initialization from Sehwag et al. (2020), $\boldsymbol{\delta}$ adopts a 0 initialization, and $\boldsymbol{\theta}$ is initialized with $\boldsymbol{\theta}_{\text{pre}}$ which stays frozen. Finally, the weights of found sparse subnetwork are further fine-tuned together with VP, as indicated in Equation 6. During this step, $\boldsymbol{\theta}$ is initialized with $\boldsymbol{\theta}_{\text{pre}}$, visual prompt $\boldsymbol{\delta}$ and mask $\mathbf{m}$ inherit the value of $\boldsymbol{\delta}_s$ and $\mathbf{m}_s$ from the previous stage, respectively. Note that here $\mathbf{m}$ is kept frozen. The detailed procedure of **VPNs** is summarized in the Appendix A. It is worth mentioning that such *data-model* co-design, *i.e.*, **VPNs**, presents a greatly improved efficiency in terms of searching desired high-quality subnetworks. For instance, compared to previous *model-centric* approaches, **VPNs** only needs half the epochs of HYDRA (Sehwag et al., 2020) and OMP (Han et al., 2015), while achieving even better accuracy (see Table A3).

## 4 Experiments

To evaluate the effectiveness of our prompting-driven sparsification method, we follow the most common evaluation of visual prompting, i.e., evaluating sparse models pre-trained on a large dataset (ImageNet-1K) on various visual domains. Moreover, we conduct extensive empirical experiments

including (1) Affirming the superior performance of **VPNs** over different datasets and architectures; (2) The transferability of **VPNs** across different datasets is investigated; (3) We further analyze the computational complexity of **VPNs** through the lens of time consumption, training epochs, and gradient calculating steps; (4) Our study also encompasses in-depth investigations into structured pruning algorithms; (5) Ablation studies are presented, which concentrate on the influence of different VP methods, pad sizes, and input sizes.

## 4.1 IMPLEMENTATION DETAILS

**Network and Datasets.** We use three pre-trained network architectures for our experiments – ResNet-18 (He et al., 2016), ResNet-50 (He et al., 2016), and VGG-16 (Simonyan & Zisserman, 2014), which can be downloaded from official Pytorch Model Zoo[1]. These models are pre-trained on the ImageNet-1K dataset (Deng et al., 2009). We then evaluate the effectiveness of VPNs over **eight** downstream datasets – Tiny ImageNet (Le & Yang, 2015), StanfordCars (Krause et al., 2013), OxfordPets (Parkhi et al., 2012), Food101 (Bossard et al., 2014), DTD (Cimpoi et al., 2014), Flowers102 (Nilsback & Zisserman, 2008), CIFAR10/100 (Krizhevsky et al., 2009), respectively. Further details of the datasets can be found in Table A1.

**Pruning Baselines.** We select **eight** representative state-of-the-art (SoTA) pruning algorithms as our baselines. (1) *Random Pruning* (Random) (Liu et al., 2022b) is commonly used as a basic sanity check in pruning studies. (2) *One-shot Magnitude Pruning* (OMP) removes weights with the globally smallest magnitudes (Han et al., 2015). (3) *The lottery ticket hypothesis* (LTH) iteratively prunes the 20% of remaining weights with the globally least magnitudes and rewinds model weights to their initial state. In our experiments, weights are rewound to their ImageNet-1K pre-trained weights, following the default configurations in Chen et al. (2021b). (4) *Pruning at initialization* (PaI) locates sparse subnetworks at the initialization phase by the defined salience metric. We opt for three widely-recognized methodologies: SNIP (Lee et al., 2019), GraSP (Wang et al., 2020b), and SynFlow (Tanaka et al., 2020) (5) *HYDRA* (Sehwag et al., 2020) prunes weights based on the least importance scores, which is the most important baseline as it can be seen as our method without the visual prompt design. (6) *BiP* (Zhang et al., 2022a), characterized as a SoTA pruning algorithm, formalizes the pruning process within a bi-level optimization framework.

**Training and Evaluation.** We follow the pruning baselines implementation in (Liu et al., 2022b), selecting optimal hyper-parameters for various pruning algorithms by grid search. As for visual prompts, our default VP design in **VPNs** employs a pad prompt with an input size of 224 and a pad size of 16. We also use an input size of 224 for all the baselines to ensure a fair comparison. More implementation details are in Table A2.

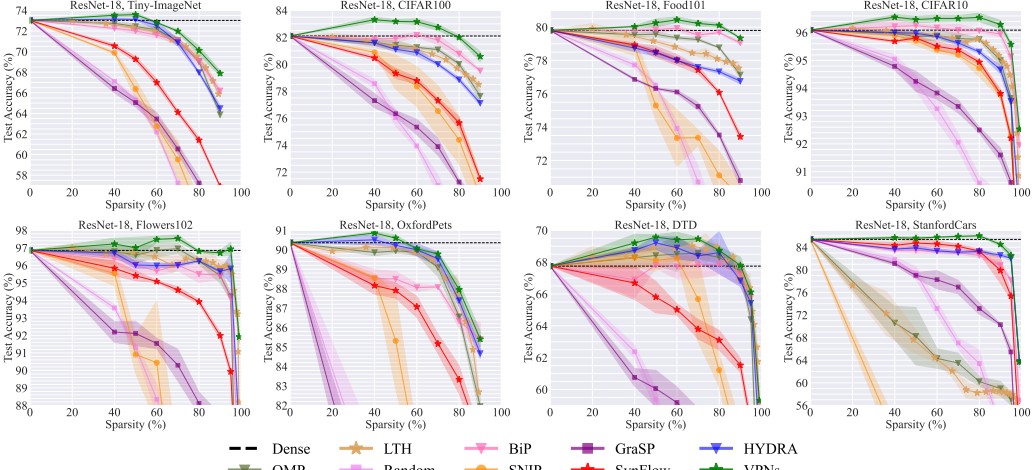

Figure 4: **Downstream Fine-tuning Results.** The performance overview of 9 unstructured pruning algorithms. All the models are pre-trained on ImageNet-1K; and then pruned and fine-tuned both on the specific downstream dataset. The performance of the dense model and VPNs' best are marked using dashed lines. All the results are averaged over 3 runs. VPNs consistently outperforms other baselines on all **eight** tasks.

---

[1]https://pytorch.org/vision/stable/models.html

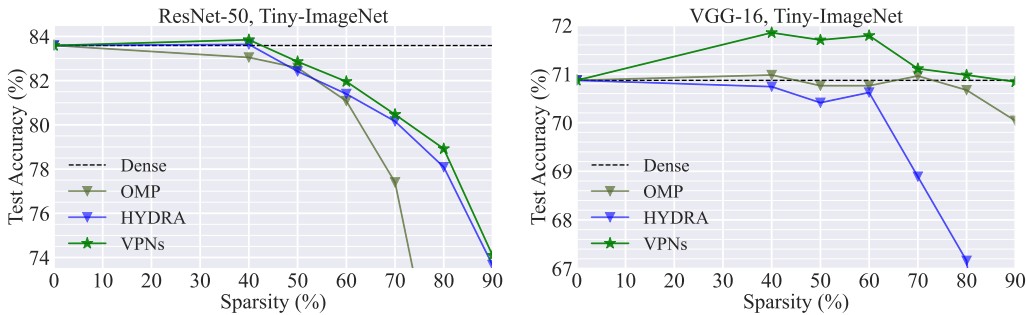

Figure 5: **Downstream Fine-tuning Results.** The performance overview of VPNs, HYDRA, and OMP. All the results are obtained with ImageNet-1K pre-trained ResNet-50 and VGG-16, fine-tuned on Tiny-ImageNet. VPNs consistently has superior performance.

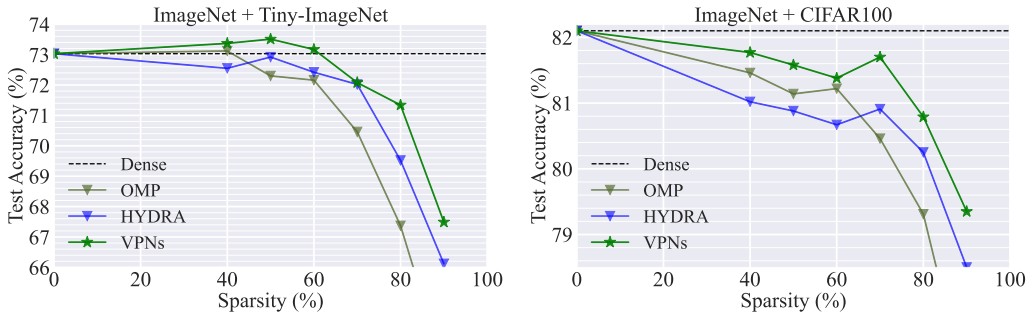

Figure 6: **ImagetNet Mask Finding and Downstream Subnetwork Tuning Results.** The performance overview of VPNs, HYDRA, and OMP. The models are pruned on ImageNet-1K and fine-tuned on Tiny-ImageNet and CIFAR100. VPNs' subnetworks consistently enjoy the best performance which indicates VPNs has transferability over datasets.

## 4.2 MAIN RESULTS

**Superior Performance of VPNs.**  Using a ResNet-18 pre-trained on ImageNet-1K, we evaluate the capability of VPNs in pruning models across multiple downstream datasets. As illustrated in Figure 4, several positive observations can be drawn: ❶ The dominance of VPNs is especially pronounced on larger datasets such as Tiny-ImageNet, CIFAR100, Food101, and CIFAR10. At 90% sparsity level, VPNs outperforms {HYDRA, BiP, LTH} by {3.41%, 1.69%, 2.00%} on Tiny-ImageNet and surpasses {HYDRA, BiP, OMP} by {3.46%, 2.06%, 2.93%} on CIFAR100. ❷ VPNs still delivers top-tier results on smaller datasets like Flowers102, OxfordPets, DTD, and Stanford-Cars. For instance, the test accuracy of VPNs is {1.12%, 2.79%, 2.71%} higher than {HYDRA, BiP, OMP} at 95% sparsity on Flowers102. ❸ VPNs outperforms fully fine-tuned dense models at high sparsity levels on all eight downstream datasets. It finds subnetworks better than dense counterparts at {50%, 70%, 80%, 90%} sparsity on {Tiny-ImageNet, CIFAR100, Food101, CIFAR10} and {70%, 50%, 90%, 90%} sparsity on {Flowers102, OxfordPets, DTD, StanfordCars}.

We conduct additional experiments with ResNet-50 and VGG-16 to investigate the performance of VPNs over different architectures. These models are pre-trained on ImageNet-1K and fine-tuned on Tiny-ImageNet. All pruning methods are applied in the fine-tuning stage. As shown in Figure 5, VPNs reaches outstanding performance across diverse architectures consistently, compared to OMP (0.85% ∼ 12.23% higher accuracy on ResNet-50) and HYDRA (1.14% ∼ 4.08% higher accuracy on VGG-16). It's noteworthy to highlight that OMP and HYDRA represent the most prominent baselines according to the results from Figure 4.

**Transferability of VPNs.**  Meanwhile, we investigate the transferability of subnetworks identified by VPNs across diverse downstream tasks. We apply VPNs, HYDRA, and OMP pruning on ResNet-18 and ImageNet-1K to identify subnetworks, subsequently fine-tune them on CIFAR100 and Tiny-ImageNet separately. The results are depicted in Figure 6, it can be observed that: ❶ VPNs consistently excels over SoTA algorithms across multiple datasets. At an 80% sparsity level on Tiny-ImageNet, VPNs has {3.97%, 1.57%} higher test accuracy than {OMP, HYDRA}. More-over, VPNs outperforms {OMP, HYDRA} by {2.75%, 0.80%} at 90% sparsity on CIFAR100. ❷ VPNs subnetworks can surpass the dense network on specific datasets. At 60% sparsity on Tiny-

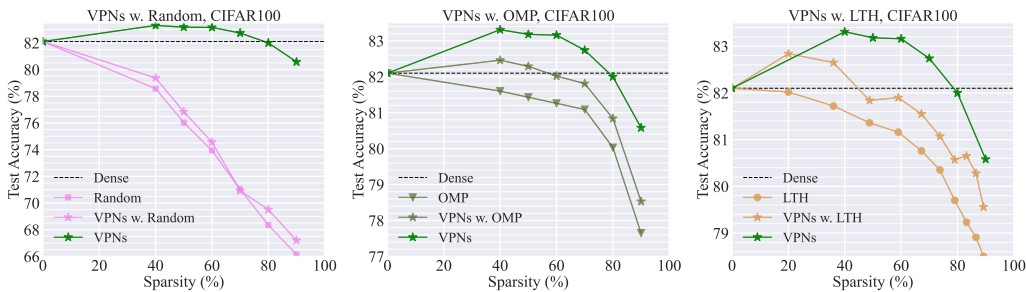

Figure 7: **VPNs Paradigm Applied to Current Methods.** The performance overview of VPNs pruning paradigm applied to Random, OMP, and LTH pruning named VPNs w. Random, VPNs w. OMP, and VPNs w. LTH. The results are based on ResNet-18 pre-trained on ImageNet-1K and fine-tuned on CIFAR100. VPNs paradigm advances Random, OMP, and LTH consistently.

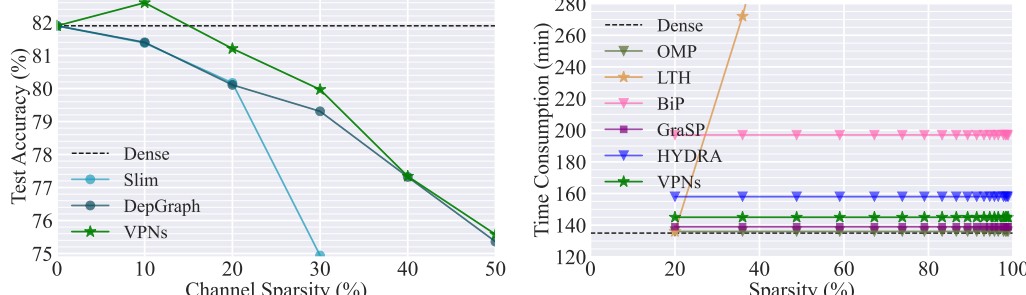

Figure 8: **Structured Pruning Results.** Test accuracy of VPNs channel-wise pruning compared to SoTA channel-wise pruning methods on ImageNet pre-trained ResNet-18 and fine-tuned on CIFAR100. VPNs structured pruning has the best performance.

Figure 9: **Time Consumption.** Time consumption of six pruning methods on ImageNet pre-trained ResNet-18 and fine-tuned on CIFAR100. VPNs is more time-efficient than BiP and HYDRA and nearly the same as GraSP.

ImageNet, subnetworks identified by VPNs have better performance than their dense counterparts. Consequently, VPNs has transferability over datasets.

**Superiority of VPNs Pruning Paradigm.** Furthermore, we endeavor to explore the potential of the VPNs pruning paradigm to enhance the effect of existing pruning algorithms. We integrate the VPNs pruning paradigm with Random, OMP, and LTH pruning, forming VPNs w. Random, VPNs w. OMP, and VPNs w. LTH respectively. For the purpose of consistency, the VP utilized in the experiment is kept identical to the one used in VPNs. The results are based on ResNet-18 pre-trained on ImageNet-1K and fine-tuned on CIFAR100. As depicted in Figure 7. We observe that VPNs combined with existing prunings consistently surpasses their original counterpart. For example, At 80% sparsity, {VPNs w. Random, VPNs w. OMP, VPNs w. LTH} surpass their corresponding original pruning by {1.16%, 0.81%, 0.79%}.

### 4.3 Additional Investigation and Ablation Study.

**VPNs for Structured Pruning.** To assess the potential of VPNs in structured pruning, we perform an empirical comparison between VPNs and renowned structured pruning techniques such as Slim (Liu et al., 2017) and DepGraph (Fang et al., 2023). The evaluations are conducted using a pre-trained ResNet-18 model on ImageNet-1K, fine-tuned on CIFAR-100. See Appendix C for more details. From the results presented in Figure 8, we observe that: ❶ VPNs enjoys superior performance consistently across various channel sparsity levels in comparison to Slim and Dep-Graph, achieving higher accuracy by $1.04\% \sim 9.54\%$ and $0.02\% \sim 1.20\%$ respectively. ❷ VPNs simultaneously reduces both training and inference FLOPs and memory costs. For example, at 10% and 20% channel sparsity levels, VPNs achieves speedup ratios of $1.1\times$ and $1.3\times$ while reducing memory costs by 15.26% and 31.71% respectively, without compromising the performance relative to the dense network. The speedup ratio is quantified as $\frac{\text{FLOPs(dense)}}{\text{FLOPs(subnetwork)}}$.

**Computational Complexity.** An effective pruning algorithm should exhibit computational efficiency. Accordingly, we evaluate the computational complexity of VPNs in comparison to the SoTA pruning methods. Our criterion contains training time consumption, training epochs, and

Figure 10: **Ablation of VP Designs.** Ablation studies of different VP designs on ImageNet pre-trained ResNet-18 and fine-tuned on CIFAR100. (a) Vary input size with pad prompt and pad size of 16. (b) Vary pad size with pad prompt and input size of 224. (c) Vary VP method with 13K prompt parameters.

gradient calculating steps with evaluations conducted on ImageNet-1K pre-trained ResNet-18 and fine-tuned on CIFAR100. Results are displayed in Figure 9 and Table A3, several positive findings can be drawn: ❶ `VPNs` consistently outperforms both BiP and HYDRA in terms of time efficiency, achieving a time reduction of $26\%$ and $8.97\%$ respectively across varying sparsity levels while exhibits a time consumption comparable to GraSP. It is also noteworthy to mention that LTH's time consumption exhibits an exponential increase in relation to sparsity growth. ❷ `VPNs` requires the fewest epochs and steps to attain optimal performance. Specifically, for achieving a $90\%$ sparsity level, `VPNs` requires $95\%$, $50\%$, and $50\%$ fewer epochs in comparison to LTH, GraSP, and HYDRA, respectively. Moreover, it demands $90\%$ and $33\%$ fewer steps than LTH and BiP separately.

**Ablation – VP Designs.** In this section, we systematically examine the impact of different VP designs on the performance of `VPNs`. Our experiments are based on ImageNet pre-trained ResNet-18 and fine-tuned on CIFAR100, where we explore various input sizes, prompt sizes, and VP methods.

*Input Size.* We employ pad prompts with a fixed pad size of 16 while varying the input size from 128 to 224 to assess the effect of input size on the performance of `VPNs`. As illustrated in Figure 10a, As the input size increases, we observe a corresponding rise in test accuracy. This underscores the imperative of harnessing the entirety of information available in the original images.

*Pad Size.* Similarly, to investigate the impact of the pad size, we fix the input size of 224 and vary the pad size of the VP from 16 to 64. The results are shown in 10b. Pad sizes 16 and 32 exhibit the best performance and the test accuracy declines as the pad sizes increase further, which indicates that a small number of prompt parameters benefits more to `VPNs` pruning performance.

*Visual Prompt Strategies.* We conduct an investigation into three distinct types of VP methods: the pad prompt, the random prompt, and the fix prompt. Figure 3 provides a visual representation of the pad prompt. In contrast, the random prompt is tunable within a randomly chosen square section of the perturbation $\delta$ as defined in Equation 2. The fix prompt, on the other hand, restricts tunability to the top-left square segment of $\delta$. See Appendix B for more details. In our experiments, all VP methods are kept consistent with 13K tunable parameters. The results are shown in Figure 10c. We observe that the pad prompt outperforms both the fix and random prompts for `VPNs`.

**Ablation – VP for Mask Finding/Subnetwork Tuning.** To assess the influence of VP during the processes of mask finding and subnetwork tuning, we conduct an ablation analysis. In this study, the VP is deactivated at specific stages of `VPNs`, resulting in two distinct algorithmic variants: " VP for Mask Finding" and "VP for Subnetwork Tuning". The results, using ResNet-18 pre-trained on ImageNet-1K on fine-tuned on CIFAR100, are depicted in Figure A11. From the results, we observe that ❶ VP for Subnetwork Tuning contributes more to the performance gains in `VPNs` than VP for Mask Finding; ❷ Inserting VP in both stages of `VPNs` achieves superior test accuracy, which suggests that our proposal effectively upgrades network sparsification.

## 5 CONCLUSION

In this work, we highlight the limitations of post-pruning prompts in enhancing vision subnetworks. To harness the potential of visual prompts for vision neural network sparsification, we introduce an innovative data-model co-design algorithm, termed **VPNs**. Comprehensive experiments across diverse datasets, architectures, and pruning methods consistently validate the superior performance and efficiency offered by **VPNs**. We further demonstrate the transferability of subnetworks identified by **VPNs** across multiple datasets, emphasizing its practical utility in a broader array of applications.

## 6 REPRODUCIBILITY STATEMENT

The authors have made an extensive effort to ensure the reproducibility of the results presented in the paper. *First*, the details of the experimental settings are provided in Section 4.1 and Appendix C. This paper investigates nine datasets and the details about each dataset are described in Table A1. The evaluation metrics are also clearly introduced in Section 4.1. *Second*, the baseline methods' implementation particulars are elucidated in Appendix C. Simultaneously, the implementation details of our method, **VPNs**, are included in Section 4.1 and Appendix C. *Third*, the codes are included in the supplementary material for further reference.

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

## A  **VPNs** ALGORITHM DETAILS

Here we provide the pseudo-code of **VPNs**. It first creates prompted images, and then locates the sparse subnetwork by jointly optimizing the mask and VP. Finally, the weights of found sparse subnetwork are further fine-tuned together with the VP.

---

**Algorithm 1 VPNs**

---

**Require:** Dataset $\mathcal{D} = \{(\mathbf{x}_1, y_1), ..., (\mathbf{x}_n, y_n)\}$, pre-trained model $f_{\boldsymbol{\theta}_{\text{pre}}}$, and sparse ratio $s$.
**Ensure:** Sparse neural network $f_{\boldsymbol{\theta}_{\text{fine-tune}} \odot \mathbf{m}_s}$.
 1: Input VP operation: $\mathbf{x}'(\boldsymbol{\delta}) = h(\mathbf{x}, \boldsymbol{\delta}) = r^i(\mathbf{x}) + \boldsymbol{\delta}^p, \mathbf{x} \in \mathcal{D}$.
 2: Sparse Initialization: Initialize importance score $\mathbf{c}$ and update the mask $\mathbf{m} = \mathbb{I}(|\mathbf{c}| > |\mathbf{c}|_{1-s})$, where $|\mathbf{c}|_{1-s}$ is the $1-s$ percentile of $|\mathbf{c}|$.
 3: **for** $i$=1 to epochs **do**
 4:     Caculate pruning loss: $\mathcal{L}_i = \mathbb{E}_{(\mathbf{x},y) \in \mathcal{D}} \mathcal{L}(f_{\boldsymbol{\theta}_{\text{pre}} \odot \mathbf{m}_{i-1}}(\mathbf{x}'(\boldsymbol{\delta}_{i-1})), y)$.
 5:     Update VP $\boldsymbol{\delta}_i$ and importance scores $\mathbf{c}_i$ via SGD calling with $\boldsymbol{\theta}$ frozen.
 6:     Update the mask: $\mathbf{m}_i = \mathbb{I}(|\mathbf{c}_i| > |\mathbf{c}_i|_{1-s})$.
 7: **end for**
 8: Re-initialization: Initialize VP with $\boldsymbol{\delta}_s$, the mask $\mathbf{m}$ with $\mathbf{m}_s$, and $\boldsymbol{\theta}$ with $\boldsymbol{\theta}_{\text{pre}}$.
 9: **for** $j$=1 to epochs **do**
10:     Caculate fine-tuning loss: $\mathcal{L}_j = \mathbb{E}_{(\mathbf{x},y) \in \mathcal{D}} \mathcal{L}(f_{\boldsymbol{\theta}_{j-1} \odot \mathbf{m}_s}(\mathbf{x}'(\boldsymbol{\delta}_{j-1})), y)$.
11:     Update VP $\boldsymbol{\delta}_j$ and model weights $\boldsymbol{\theta}_j$ via SGD calling with $\mathbf{m}$ frozen.
12: **end for**

---

## B  VISUAL PROMPT DESIGN DETAILS

We explore three different kinds of VP designs, namely **Pad Prompt**, **Random Prompt**, and **Fix Prompt** (Bahng et al., 2022). Each of these VP methods can be formulated into two steps: ❶ *Input resize and pad operation.* We resize the original image $\mathbf{x}$ to a designated input size $i \times i$ and subsequently pad it to $224 \times 224$ with 0 values to derive the resized image. This procedure is represented as $r^i(\mathbf{x})$, where $r(\cdot)$ refers to the resize and pad operation and $i$ indicates the input size. ❷ *Perturbation mask operation.* We initiate the perturbation parameters of $\boldsymbol{\delta}$ as a $224 \times 224$ matrix with a portion being masked. The input prompting operation is then formulated as Equation 2. All the VP variants have the same input resize and pad operation and the differences for them lie in the distinct masked regions during the perturbation mask operation.

*Pad Prompt.* This kind of prompt masks a central square matrix of the perturbation $\boldsymbol{\delta}$, while keeping the left four peripheral segments tunable. The width of each side is denoted as the pad size, marked as $p$. Figure 3 provides a visual representation of the pad prompt. The number of tunable prompt parameters numbers for the pad prompt is $4p(224 - p)$.

*Fix Prompt.* This prompt design retains the top-left square segment of the perturbation $\boldsymbol{\delta}$ tunable, masking the remaining areas of $\boldsymbol{\delta}$. The width of the tunable square is denoted prompt size, marked as $p$. The number of tunable prompt parameters for the fix prompt is $p^2$.

*Random Prompt.* The random prompt keeps a random square segment of the perturbation $\boldsymbol{\delta}$ tunable, masking other areas of $\boldsymbol{\delta}$ during each forward pass. Similarly, the width of the tunable square is denoted as $p$ and referred to as the prompt size. The random prompt has a $p^2$ parameter number.

Table A1: Datasets configurations.

| Dataset | Train Set Size | Test Set Size | Class Number | Batch Size |
|---|---|---|---|---|
| Flowers102 | 5726 | 2463 | 102 | 128 |
| DTD | 3760 | 1880 | 47 | 64 |
| Food101 | 75750 | 25250 | 101 | 256 |
| OxfordPets | 3680 | 3669 | 37 | 64 |
| StanfordCars | 8144 | 8041 | 196 | 128 |
| CIFAR10 | 50000 | 10000 | 10 | 256 |
| CIFAR100 | 50000 | 10000 | 100 | 256 |
| Tiny ImageNet | 100000 | 10000 | 200 | 256 |
| ImageNet | 1281167 | 50000 | 1000 | 1024 |

Table A2: Configurations for unstructured pruning. **m** indicates hyperparameters for maskfinding and $\theta$ represents hyperparameters for weight tuning.

| Method | Epochs | Optimizer | Initial Learning Rate | Learning Rate Decay | Weight Decay |
|---|---|---|---|---|---|
| Random | 120 | SGD | 0.01 | Cosine Decay | 0.0001 |
| OMP | 120 | SGD | 0.01 | Cosine Decay | 0.0001 |
| LTH | 120 | SGD | 0.01 | Cosine Decay | 0.0001 |
| SNIP | 120 | SGD | 0.01 | Cosine Decay | 0.0001 |
| GraSP | 120 | SGD | 0.01 | Cosine Decay | 0.0001 |
| SynFlow | 120 | SGD | 0.01 | Cosine Decay | 0.0001 |
| BiP | 60 | Adam for $\delta$, SGD for $\theta$ | 0.0001 for $\delta$, 0.01 for $\theta$ | Cosine Decay | 0.0001 |
| HYDRA | 60 for $\delta$, 60 for $\theta$ | Adam for $\delta$, SGD for $\theta$ | 0.0001 for $\delta$, 0.01 for $\theta$ | Cosine Decay | 0.0001 |
| **VPNs** | 30 for $\delta$, 30 for $\theta$ | Adam for $\delta$, SGD for $\theta$ | 0.0001 for $\delta$, 0.01 for $\theta$ | Cosine Decay | 0.0001 |

Table A3: **Training Epochs and Steps.** Training epochs and gradients calculating steps among different pruning algorithms on ImageNet pre-trained ResNet-18 and fine-tuned on CIFAR100. VPNs takes the least epochs and steps to obtain the superior performances of CIFAR100 in Figure 4.

| Method | Epochs | | | | Steps | | | |
|---|---|---|---|---|---|---|---|---|
| Sparsity | 20% | 59% | 89.26% | 95.60% | 20% | 59% | 89.26% | 95.60% |
| LTH | 120 | 480 | 1200 | 1680 | 23520 | 94080 | 235200 | 329280 |
| OMP | | | 120 | | | | 23520 | |
| GraSP | | | 120 | | | | 23523 | |
| BiP | | | 60 | | | | 34560 | |
| HYDRA | 60 mask finding + 60 subnetwork tuning | | | | 11760 mask finding + 11760 subnetwork tuning | | | |
| **VPNs** | 30 mask finding + 30 subnetwork tuning | | | | 11760 mask finding + 11760 subnetwork tuning | | | |

## C  IMPLEMENTATION DETAILS

**Datasets.** We use the standard train-test division of 9 image classification datasets to implement our method and report the test set accuracy. All images are resized to $224 \times 224$ in mask finding and weight tuning processes. The configurations of the datasets are summarized in Table A1.

**Hyperparameters for Unstructured Pruning.** For Random (Liu et al., 2022b), OMP (Han et al., 2015), LTH (Chen et al., 2021b), SNIP (Lee et al., 2019), GraSP (Wang et al., 2020b), SynFlow (Tanaka et al., 2020), we use SGD optimizer with a learning rate of 0.01 and cosine decay scheduler. For HYDRA (Sehwag et al., 2020), BiP (Zhang et al., 2022a) and VPNs, we use Adam optimizer with a learning rate of 0.0001 and cosine decay scheduler for mask finding and SGD optimizer with a learning rate of 0.01 and cosine decay scheduler for weight tuning. Further details regarding hyperparameter configurations can be found in Table A2. In this work, we use **m** to indicate hyperparameters for mask finding and $\theta$ represents hyperparameters for weight tuning.

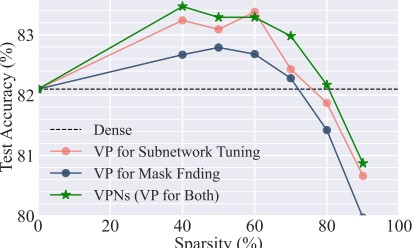

Figure A11: **Ablation of VP for Mask Finding/Subnetwork Tuning.** Ablation of VPNs only using VP in mask finding or subnetwork tuning on ImageNet pre-trained ResNet-18 and fine-tuned on CIFAR100.

**Hyperparameters for Structured Pruning.** We follow the implementation in Fang et al. (2023) to reproduce the results of Slim (Liu et al., 2017) and DepGraph (Fang et al., 2023). For the structured pruning version of **VPNs**, we warm up 5 epochs before pruning. In the mask finding stage, we train 30 epochs using the Adam optimizer with a learning rate of 1 and cosine decay scheduler. The weight decay is set to 0.01. In the weight tuning stage, we train 30 epochs using the SGD optimizer with a learning rate of 0.01, cosine decay scheduler, and weight decay of 0.0001.

## D  ADDITIONAL RESULTS

**Computational Complexity.** Here we provide additional results of computational complexity analysis among **VPNs** and our baselines through the lens of training epochs and gradient calculating steps. The experiments are conducted on ImageNet pre-trained ResNet-18 and fine-tuned on

CIFAR100. From Table A3, we observe that **VPNs** requires the fewest epochs and steps to attain optimal performance, which means **VPNs** is highly computationally efficient.

**Ablation – VP for Mask Finding/Subnetwork Tuning.** Figure A11 shows the ablation results among **VPNs** ("VP for Both"), " VP for Mask Finding" and "VP for Subnetwork Tuning". The experiments are based on ResNet-18 pre-trained on ImageNet-1K on fine-tuned on CIFAR100. We find that **VPNs** achieves superior performance, which means inserting VP in both stages is the best.

**Additional Experiments of Baselines with VPNs Pruning Paradigm.** We conduct supplemental experiments by applying **VPNs** on some of the best baselines such as LTH and OMP named **VPNs** w. LTH and **VPNs** w. OMP on ImageNet pre-trained ResNet-18 network and Tiny-ImageNet. **VPNs** w. LTH and **VPNs** w. OMP use the same VP and joint optimization method as **VPNs**. We again observe that our method is superior to baselines with VP as displayed in Table A4, which indicates our method is better than baselines with VP.

Table A4: **More results of Baselines with VPNs Pruning Paradigm.** Performance comparison of our method, **VPNs** w. LTH, and **VPNs** w. OMP on ImageNet pre-trained ResNet-18 network and Tiny-imageNet.

| Method | Sparsity | |
|---|---|---|
| | 50% | 90% |
| **VPNs** w. LTH | 73.71 | 67.03 |
| **VPNs** w. OMP | 73.56 | 64.78 |
| **VPNs** | 73.82 | 67.89 |

**Results on ImageNet.** We carry out additional experiments on ImageNet using our method and some of the best baselines such as HYDRA and OMP on ImageNet pre-trained ResNet-18 network. The empirical results are shown in Table A5, our method has $\{0.52\%, 2.87\%\}$ higher accuracy than $\{\text{HYDRA, OMP}\}$ at 90% sparsity, which illustrates the effectiveness of our method on ImageNet.

Table A5: **Results on ImageNet.** Performance comparison of our method, HYDRA, and OMP on ImageNet pre-trained ResNet-18 network and ImageNet.

| Method | Sparsity | |
|---|---|---|
| | 50% | 90% |
| HYDRA | 68.91 | 66.62 |
| OMP | 69.31 | 64.27 |
| **VPNs** | 69.47 | 67.14 |

**Results on Object Detection Tasks.** We implement additional experiments on Pascal VOC 2007 (Everingham et al., 2010), which is a widely used dataset for object detection tasks. We compare our method to HYDRA and OMP on YOLOv4 (Bochkovskiy et al., 2020) using ImageNet pre-trained ResNet-18 as the backbone. Our method achieves $\{3.78\%, 2.67\%\}$ higher AP than $\{\text{HYDRA, OMP}\}$ at 90% sparsity level as presented in Table A6, which demonstrates the superiority of our method on object detection.

**Results of Baselines Learning Additional Parameters.** We implement supplemental experiments on ImageNet pre-trained ResNet-18 on CIFAR100, using our method, HYDRA and OMP. HYDRA and OMP learn an additional 13k parameters (the number of parameters in the VP of **VPNs**) than **VPNs**. We find that our method obtains $\{3.46\%, 3.06\%\}$ higher accuracy than $\{\text{HYDRA,}$

Table A6: **Results on Pascal VOC** 2007. AP comparison of our method, HYDRA, and OMP on YOLOv4 with ImageNet pre-trained ResNet-18 backbone and Pascal VOC 2007.

| Method | Sparsity | |
|--------|-----|-----|
| | 50% | 90% |
| HYDRA | 35.25 | 32.74 |
| OMP | 35.01 | 33.85 |
| **VPNs** | 38.37 | 36.52 |

Table A7: **Results of Baselines Learning Additional Parameters.** Performance comparison of our method, HYDRA, and OMP on ImageNet pre-trained ResNet-18 and CIFAR100. HYDRA and OMP learn 13k additional parameters than our method.

| Method | Sparsity | |
|--------|-----|-----|
| | 50% | 90% |
| HYDRA | 81.03 | 77.12 |
| OMP | 81.36 | 77.52 |
| **VPNs** | 83.18 | 80.58 |

OMP} at 90% sparsity level as displayed in Table A7, which manifests that our method is better than the baselines with additional parameters.

**Comparison to Baselines using more data augmentations.** To compare the effect of visual prompting in **VPNs** pruning paradigm to data augmentations. We contrast our method without mix-ups with HYDRA and OMP with mix-ups on ImageNet pre-trained ResNet-18 network and CIFAR10 dataset. Our method achieves $\{5.46\%, 3.50\%\}$ higher accuracy than {HYDRA, OMP} at 90% sparsity level as shown in Table A8, which indicates that the **VPNs** pruning paradigm is better than baselines using more data augmentations.

Table A8: **Results of Baselines using more data augmentations.** Performance comparison of our method without mix-ups to HYDRA and OMP with mix-ups on ImageNet pre-trained ResNet-18 and CIFAR10.

| Method | Sparsity | |
|--------|-----|-----|
| | 50% | 90% |
| HYDRA | 92.72 | 90.83 |
| OMP | 94.63 | 92.79 |
| **VPNs** | 96.47 | 96.29 |

**Latency and FLOPs for Structured Pruning.** We provide the latency and FLOPs of **VPNs** structured pruning on ResNet-18 pre-trained on ImageNet-1K and fine-tuned on CIFAR100 in Table A9. Our method achieves latency speedup ratios of $\{1.1\times, 1.2\times\}$ and FLOPs speedup ratios of $\{1.1\times, 1.3\times\}$ at $\{10\%, 20\%\}$ channel-wise sparsity without compromising the performance relative to the dense network, as indicated in Figure 8.

Table A9: **Latency and FLOPs.** The latency and FLOPs of **VPNs** structured pruning on ImageNet pre-trained ResNet-18 and CIFAR100.

| Sparsity | Latency(ms) | FLOPs(G) |
|----------|-------------|----------|
| Dense | $2.23 \pm 0.03$ | 1.82 |
| 10% sparsity | $2.02 \pm 0.02$ | 1.68 |
| 20% sparsity | $1.86 \pm 0.02$ | 1.45 |

**More Results on Structured Pruning.**    To further demonstrate the superior performance of our method in structured pruning, we conduct additional experiments on ImageNet pre-trained ResNet-18 and CIFAR100 using GReg (Wang et al., 2020a) and LAMP (Lee et al., 2020), which are two recent structured pruning methods. We observe that our method still maintains the best performance as shown in Table A10, which again indicates the superiority of our method in structured pruning.

Table A10: **Results on Structured Pruning.** Performance comparison of our method, GReg, and LAMP on ImageNet pre-trained ResNet-18 and CIFAR100.

| Method | Sparsity | |
|---|---|---|
| | 20% | 50% |
| GReg | 78.91 | 62.63 |
| LAMP | 80.7 | 75.46 |
| **VPNs** | 81.21 | 75.58 |

**Results of Baselines Using Lower Resolution.**    We conduct additional experiments on ImageNet pre-trained ResNet-18 and CIFAR100 using some of the best baselines such as HYDRA and OMP, which use an input resolution of 192 and prune 13k fewer parameters than our method. We find that our method achieves $\{3.94\%, 3.69\%\}$ higher accuracy than $\{$HYDRA, OMP$\}$ at 90% sparsity level from the results displayed in Table A11, which indicates our method is superior to baselines using lower resolution and pruning fewer parameters.

Table A11: **Results of Baselines Using Lower Resolution.** Performance comparison of our method, HYDRA, and OMP on ImageNet pre-trained ResNet-18 and CIFAR100. HYDRA and OMP use 192 as the input resolution and prune 13k fewer parameters than our method.

| Method | Sparsity | |
|---|---|---|
| | 50% | 90% |
| HYDRA | 80.9 | 76.64 |
| OMP | 80.89 | 76.89 |
| **VPNs** | 83.18 | 80.58 |

**Comparision on Smaller Models.**    We conduct additional experiments on ImageNet pre-trained MobileNet (Sandler et al., 2018) and CIFAR100 using our method, HYDRA, and OMP. We observe that our method achieves $\{2.24\%, 1.27\%\}$ higher accuracy than $\{$HYDRA, OMP$\}$ at 50% sparsity level as shown in Table A12, which indicates the superiority of our method on small networks.

Table A12: **Results on Small Networks.** Performance comparison of our method, HYDRA, and OMP on ImageNet pre-trained MobileNet and CIFAR100.

| Method | Sparsity | |
|---|---|---|
| | 50% | 90% |
| HYDRA | 79.12 | 70.02 |
| OMP | 80.09 | 41.82 |
| **VPNs** | 81.36 | 75.5 |

**Results of Training from Scratch.**    To explore whether visual prompting upgrades models' sparse training in the setting of training from scratch, we conduct additional experiments by applying the **VPNs** pruning paradigm on SynFlow named **VPNs** w. SynFlow on ResNet-18 and CIFAR100. We

choose SynFlow because it is a PaI pruning method independent of model weights, while **VPNs** is dependent on model weights and the random weights make the mask-finding stage of **VPNs** useless. The empirical results are shown in Table A13, we observe that **VPNs** w. SynFlow achieves $\{8.85\%, 5.91\%\}$ higher accuracy than the original SynFlow at $50\%$ and $90\%$ sparsity levels, which indicates visual prompting significantly enhances models' sparsification in the setting of training from scratch.

Table A13: **Results of Training from Scratch.** Performance comparison of **VPNs** w. SynFlow and SynFlow pruning from scratch on ResNet-18 and CIFAR100.

| Method | Sparsity | |
|---|---|---|
| | 50% | 90% |
| SynFlow | 66.77 | 64.46 |
| **VPNs** w. SynFlow | 75.62 | 70.37 |

**Comparison with BiP on ImageNet.**    To further explore the superiority of our method on ImageNet, we conducted additional experiments by applying **VPNs** and BiP on ImageNet pre-trained ResNet-18 and ImageNet. We observe that our method outperforms BiP at $20\%$ and $70\%$ sparsity levels as indicated in Table A14, which demonstrates the superiority of our method over BiP.

Table A14: **Comparison with BiP on ImageNet.** Performance comparison of **VPNs** and BiP on ImageNet pre-trained ResNet-18 network and ImageNet dataset.

| Method | Sparsity | |
|---|---|---|
| | 20% | 70% |
| BiP | 69.37 | 68.85 |
| **VPNs** | 69.49 | 69.13 |

