# OpenReview forum: "Visual Prompting Upgrades Neural Network Sparsification: A Data-Model Perspective"
_ICLR.cc/2024/Conference — Submitted to ICLR 2024_

### Official Review · Reviewer_hiB3 · 2023-10-21

**Soundness:** 3 good
**Presentation:** 3 good
**Contribution:** 2 fair
**Rating:** 5
**Confidence:** 4

**Summary:**

To upgrade vision model sparsification, the paper proposed a data-model co-design sparsification paradigm, where integrating input image with the learnable perturbation, and a network tuning strategy is designed to optimize this issue.

The algorithm has demonstrated excellent performance on CIFAR-10 and CIFAR-100 datasets.

**Strengths:**

1. The manuscript exhibits a commendable level of writing proficiency, featuring well-crafted graphics that enhance the overall presentation and a compelling narrative.

2. The algorithm has showcased remarkable efficacy when applied to the CIFAR-10 and CIFAR-100 datasets, achieving good performance.

**Weaknesses:**

1. **Inadequate experiments.** This is a primary concern for the reviewer. The paper only presents experiments on CIFAR-10 and CIFAR-100, which, in the era of big data, are considered insufficient. These experiments do not adequately demonstrate the performance of the proposed method. Conducting experiments on larger datasets, such as ImageNet-1k, is essential.

Additionally, it's worth noting that the method utilizes ImageNet-1K pre-trained weights, which were trained on a resolution of 224. However, the method is tested on CIFAR data with a resolution of 32. It is evident that padding the data to a resolution of 224 can significantly boost performance. From this perspective, experiments specifically conducted on ImageNet with a resolution of 224, and direct performance comparisons with fine-tuning on this resolution, are crucial.

Figure 10 further substantiates this conclusion, showing that the optimal performance is achieved at a resolution of 224, with diminishing performance as the resolution decreases. Therefore, padding CIFAR data with a resolution of 32 to 224 doesn't necessarily demonstrate the superiority of the method. The gains observed in this case could be attributed to the ImageNet-1K pre-trained weights at a resolution of 224.

2. The reviewer also suggests providing performance comparisons with smaller models, as performance metrics on sparser models, such as using MobileNet, would be more indicative and informative.


3. Furthermore, the method exhibits significant limitations, as it necessitates the use of pre-trained weights from a larger dataset. The current version seems to require transforming the ImageNet model to CIFAR. It would be insightful to explore the performance without pre-trained weights. Additionally, conducting comparisons on a larger pre-trained dataset, such as ImageNet, appears necessary for the current version.

**Questions:**

In each figure, the authors have plotted curves labeled as "our best," which may not be entirely necessary as this information can be inferred from the VPNs curve.

Moreover, this plotting style has the potential to cause confusion; upon initial review, it might be perplexing why the curve representing "ours" appears as a straight line.

---

> ### Author Response · Authors · 2023-11-18
> **Point-to-point Response to Reviewer hiB3 (Part 1/2)**
>
> Many thanks to reviewer hiB3 for acknowledging that our method achieves “remarkable efficiency” on CIFAR10 and CIFAR100, our writing proficiency is  “commendable”, our graphics are “well-crafted”, and our narrative is “compelling”. We sincerely appreciate all constructive suggestions, which help us to improve our paper further. To address reviewer hiB3’s concerns, we provide pointwise responses below.
>
> **[Cons 1. Only presenting experiments on CIFAR-10 and CIFAR-100 and conducting experiments on large datasets is essential.]**
>
> This is a possible misunderstanding of our experiment setting. As we state in the Abstract, Line 15; Section 4.1, Paragraph 1; Section 4.2, Paragraph 1; [Figure 4](https://imgur.com/YF7ODNx), we conduct our experiments on eight datasets which are Tiny-ImageNet, Food101, CIFAR100, CIFAR10, DTD, Flowers102, StanfordCars, and OxfordPets. We also display the results of pruning on ImageNet and fine-tuning on Tiny-ImageNet and CIFAR100 in [Figure 6](https://imgur.com/pOVS2DX).
>
> **Additional experiments on ImageNet-1K.** To further alleviate reviewer hiB3’s concern, we conducted additional experiments on ImageNet-1K using our method and some of the best baselines such as HYDRA and OMP on ImageNet-1K pre-trained ResNet-18 network. The empirical results are shown in Table R10, **our method achieves the best accuracy** at 50% and 90% sparsity levels, which illustrates the effectiveness of our method on ImageNet. We also report the results in the revision of our paper in Table A5. Due to the time limitation of rebuttal, we provide outcomes only at two sparsity levels, more sparsity levels are promised in our final version.
>
> Table R10. Performance comparison of our method, HYDRA, and OMP on ImageNet pre-trained ResNet-18 and ImageNet.
> | Method | 50% sparsity | 90% sparsity |
> | :----------: | :----------: | :----------: |
> | HYDRA | 68.91 | 66.62 |
> | OMP | 69.31 | 64.27 |
> | VPNs (ours) | 69.47 | 67.14 |
>
> **[Cons 2. Our method uses a resolution of 32.]**
>
> Thanks for pointing out the question. We are aware of the role of resolution in the experiments and our responses to this concern are as follows.
>
> 1. [We use a resolution of 224]  We kindly indicate that we unify the resolution of 224 in our method and the baselines in all of our results in Figure [4](https://imgur.com/YF7ODNx) \ [5](https://imgur.com/7WL9Q9v) \ [6](https://imgur.com/pOVS2DX) \ [7](https://imgur.com/zqmXSrW) \ [8](https://imgur.com/VkoqArm). As displayed in Section 4.1, Paragraph 3, Line 5, we use an input size of 224 and pad size of 16, which corresponds to [Figure 3](https://imgur.com/y1fRg7v) that i = 224 and p = 16. This is also illustrated in our code, the main.py file, line 30, we use an input size of 224. For reviewer hiB3’s concern, we significantly polished our paper in Section 4.1, Paragraph 3 in our revision by specifying both our method and baselines using a resolution of 224 to avoid misunderstanding.
>
> 2. [We have results on datasets that have a resolution of 224] We gently state that we present results on Tiny-ImageNet which has an original resolution of 224 in Figure [4](https://imgur.com/YF7ODNx) \ [5](https://imgur.com/7WL9Q9v) \ [6](https://imgur.com/pOVS2DX). The findings of these experiments also demonstrate our method is superior to all the baselines at an original resolution of 224 circumstances.
>
> 3. [Additional results on ImageNet] The results on ImageNet in Table R10 also illustrate that our method achieves superior performance on ImageNet.

---

> ### Author Response · Authors · 2023-11-18
> **Point-to-point Response to Reviewer hiB3 (Part 2/2)**
>
> **[Cons 3. Providing performance comparisons with smaller models such as MobileNet.]**
>
> Thank you for the great suggestion. We conducted additional experiments on ImageNet pre-trained MobileNet and CIFAR100 using our method, HYDRA, and OMP. **We observe that our method achieves {2.24%, 1.27%} higher accuracy** than {HYDRA, OMP} at 50% sparsity level as shown in Table R11. The results are also included in the revision of our paper in Table A12. Due to the time limitation, we provide outcomes only at two sparsity levels, more sparsity levels are promised in our final version.
>
> Table R11. Performance comparison of our method, HYDRA, and OMP on ImageNet pre-trained MobileNet and CIFAR100.
> | Method | 50% sparsity | 90% sparsity |
> | :----------: | :----------: | :----------: |
> | HYDRA | 79.12 | 70.02 |
> | OMP | 80.09 | 41.82 |
> | VPNs (ours) | 81.36 | 75.5 |
>
> **[Cons 4. It would be insightful to explore the performance without pre-trained weights]**
>
> Thank you for the advice. We state that the pruning paradigm of our method which requires first finding masks in a trained model and then tuning the subnetwork to recover the accuracy can’t be used on models without pre-training. This is because a model without pre-training has random weights making the mask-finding stage of our method useless.
>
> **Additional experiments of training from scratch.** To further alleviate reviewer hiB3’s concern, we conducted additional experiments by applying the VPNs pruning paradigm on SynFlow named VPNs w. SynFlow and pruning from scratch on ResNet-18 and CIFAR100. We observe that **VPNs w. SynFlow(with VP) achieves {8.85%, 5.91%} higher accuracy** than the original SynFlow (without VP) at {50%, 90%} sparsity levels as shown in Table R12, which indicates visual prompting significantly enhances models’ sparsification in the setting of training from scratch. We also report the results in the revision of our paper in Table A13. Due to the time limitation, we provide outcomes only at two sparsity levels, more sparsity levels are promised in our final version.
>
> Table R12. Performance comparison of VPNs w. SynFlow and SynFlow pruning from scratch on ResNet-18 and CIFAR100.
> | Method | 50% sparsity | 90% sparsity |
> | :----------: | :----------: | :----------: |
> | SynFlow (without VP) | 66.77% | 64.46% |
> | VPNs w. SynFlow (with VP) | 75.62% | 70.37% |
>
> **[Cons 5. It might be perplexing why the curve representing "ours" appears as a straight line.]**
>
> Thank you for the great point. We use a similar plot format as in [15]. Regarding reviewer hiB3’s concern, we deleted the “Our Best” dashed lines in all plots in the revision of our paper.
>
> [15] Yihua Zhang, Yuguang Yao, Parikshit Ram, Pu Zhao, Tianlong Chen, Mingyi Hong, Yanzhi Wang, and Sijia Liu. Advancing model pruning via bi-level optimization. Advances in Neural Information Processing Systems, 35:18309–18326, 2022a.

---

> ### Author Response · Authors · 2023-11-20
> **Response to Reviewer hiB3**
>
> Dear Reviewer **hiB3**,
>
> We thank reviewer **hiB3** time for the review and constructive comments. We really hope to have a further discussion with the reviewer **hiB3** to see if our response solves the concerns.
>
> In our response, we have (1) clarified the resolution of our method; (2) conducted additional experiments on ImageNet, MobileNet, and models without pre-trained weights, further indicating the superiority of our method.
>
> We genuinely hope reviewer **hiB3** could kindly check our response. Thanks!
>
> Best wishes,
>
> Authors

---

> > ### Comment · Reviewer_hiB3 · 2023-11-21
> > **Response to authors**
> >
> > Thank you for the authors' response.
> >
> > Firstly, a suggestion is to avoid using imgur.com as it seems that many images are now expired. If you make any changes, please include them in the updated draft and indicate in your response.
> >
> > This would provide a more straightforward way to align and review the modifications.
> >
> > Overall, the authors have addressed the majority of my concerns.  The reviewer suggests supplementing additional sparsity levels on ImageNet, such as 20% and 70%, as it appears to be more informative and valuable for reference.
> >
> > However, there is still a concern raised by the reviewer: in the experiments on ImageNet, the authors compared HYDRA and OMP, both of which were published before 2020. This does not make sense, and the authors should consider comparing with more recent and stronger works, such as BiP [1].
> >
> > For the experiments on ImageNet with ResNet-18, BiP seems to achieve around 71% and 70% performance at 50% and 90% sparsity (Figure A7 of the paper).
> >
> > [1] Zhang, Yihua, Yuguang Yao, Parikshit Ram, Pu Zhao, Tianlong Chen, Mingyi Hong, Yanzhi Wang, and Sijia Liu. "Advancing model pruning via bi-level optimization." Advances in Neural Information Processing Systems 35 (2022): 18309-18326.

---

> > > ### Author Response · Authors · 2023-11-22
> > > **Point-to-point Response to Reviewer hiB3 (Part 1/2)**
> > >
> > > Many thanks to reviewer hiB3 for your response. We are happy to hear that most of your concerns have been addressed. Regarding your new questions, we provide pointwise responses below.
> > >
> > > **[Cons 1. Including changes in the updated draft and indicate in the response.]**
> > >
> > > Thank you for the suggestion. We apologize for any inconvenience. We have included all changes in the revision. To further clarify the changes in the revision, we reiterate the changes here:
> > >
> > > - We deleted the “Our Best” dashed line in all Figures in the revision.
> > >
> > > - The hyperlink for Figure 4 corresponds to Figure 4 in the revision, which indicates our method is superior to eight baselines on eight datasets.
> > >
> > > - The hyperlink for Figure 5 corresponds to Figure 5 in the revision, which indicates our method is superior to the best baselines on ResNet-50 and VGG-16.
> > >
> > > - The hyperlink for Figure 6 corresponds to Figure 6 in the revision, which indicates the subnetworks found by our method on ImageNet are superior to those found by baselines.
> > >
> > > - The hyperlink for Figure 7 corresponds to Figure 7 in the revision, which indicates visual prompting upgrades the baselines.
> > >
> > > - The hyperlink for Figure 8 corresponds to Figure 8 in the revision, which indicates our method is better than the state-of-the-art structured pruning.
> > >
> > > - We add the results in Table R10 in Table A5 in the revision, which demonstrates the superiority of our method on ImageNet.
> > >
> > > - We add the results in Table R11 in Table A12 in the revision, which illustrates the superiority of our method on MobileNet.
> > >
> > > - We add the results in Table R12 in Table A13 in the revision, which indicates the effectiveness of visual prompting in models without pre-trained weights.
> > >
> > >
> > > **[Cons 2. Compared with more recent works such as BiP at 20% and 70% sparsity levels on ImageNet.]**
> > >
> > > Thank you for the advice. We agree that more recent work BiP is indeed a strong baseline. In the meantime, we believe HYDRA and OMP are also very significant in our work for the following reasons:
> > >
> > > - HYDRA is the most important baseline as indicated in Section 4.1, Paragraph 2, Line 11 in the revision.
> > >
> > > - HYDRA and OMP achieve the second/third best performance (only lower than our method) on Tiny-ImageNet, DTD, Flowers102, StanfordCars, and OxfordPets as shown in Figure 4 in our paper.
> > >
> > > Regarding the reviewer hiB3’s suggestions on including more comparisons with BiP, we kindly indicate that we used BiP as one of the baselines and demonstrated the superiority of our method on eight datasets which are Tiny-ImageNet, Food101, CIFAR100, CIFAR10, DTD, Flowers102, StanfordCars, and OxfordPets in Figure 4 in our paper.
> > >
> > > To further alleviate reviewer hiB3’s concern, we conducted additional experiments on ImageNet pre-trained ResNet-18 and ImageNet using our method and BiP at 20% and 70% sparsity levels. We observe that **our method outperforms BiP** as illustrated in Table R15, demonstrating the superiority of our method on ImageNet. We also added the results in Table A14 in the revision.
> > >
> > > Table R15. Performance comparison of our method and BiP on ImageNet pre-trained ResNet-18 network and ImageNet dataset.
> > > | Method | 20% sparsity | 70% sparsity |
> > > | :----------: | :----------: | :----------: |
> > > | BiP | 69.37 | 68.85 |
> > > | VPNs (ours) | 69.49 | 69.13 |

---

> > > ### Author Response · Authors · 2023-11-22
> > > **Point-to-point Response to Reviewer hiB3 (Part 2/2)**
> > >
> > > **[Cons 3. BiP achieves around 71% and 70% performance at 50% and 90% sparsity as indicated in [1].]**
> > >
> > > We would like to emphasize the performance differences between our paper and [1] below:
> > >
> > > - **[1] and our paper use a different setting.** The methods in [1] are trained from scratch, while all of our methods are based on ImageNet pre-trained model weights officially released by Pytorch [2].
> > >
> > > - **[1] and our paper use different dense models.** Following the statement above, we consistently chose the ImageNet pre-trained weights officially released by Pytorch [2]. In contrast, the dense models used by BiP in [1] trained by themselves achieve an accuracy of nearly 70.9% compared to 67.39% from Pytorch. Therefore, directly comparing to the numbers in [1] might result in a disadvantage against our setting. Since [1] didn’t provide their dense model’s weights, we further compared the performance increase with respect to the dense network. **Our method achieves a higher performance increase** compared to BiP in [1] as shown in Table R16. We have also emailed the BiP team for the improved checkpoint. If they provide a better checkpoint later, we will produce new results on top of it in our final version.
> > >
> > > Table R16. Comparison of performance increase on ImageNet between our paper and [1]. Our method achieves a higher performance increase.
> > > | Method | Dense | Sparsity | Pruned | Delta (Pruned - Dense) |
> > > | :----------: | :----------: | :----------: | :----------: | :----------: |
> > > | BiP - [1] | 70.9 | 50% | 71.4 | 0.5 |
> > > | BiP - [1] | 70.9 | 90% | 70.4 | -0.5 |
> > > | VPNs (ours) | 67.39 | 50% | 69.47 | 2.08 |
> > > | VPNs (ours) | 67.39 | 90% | 67.14 | -0.25 |
> > >
> > > [1] Zhang, Yihua, Yuguang Yao, Parikshit Ram, Pu Zhao, Tianlong Chen, Mingyi Hong, Yanzhi Wang, and Sijia Liu. "Advancing model pruning via bi-level optimization." Advances in Neural Information Processing Systems 35 (2022): 18309-18326.
> > >
> > > [2] https://pytorch.org/vision/stable/models.html

---

### Official Review · Reviewer_eRqj · 2023-10-31

**Soundness:** 3 good
**Presentation:** 4 excellent
**Contribution:** 2 fair
**Rating:** 3
**Confidence:** 5

**Summary:**

This paper proposes a joint model pruning and visual prompting learning method. By combining these two methods, it could recover the performance loss by pruning. This method is validated on several pruning methods and datasets to demonstate its universality.

**Strengths:**

The idea is easy to follow and effective.It is interesting to see that only a small number of learned parameters could improve the performance.

**Weaknesses:**

1. I think the authors should focus more on the structured sparse case, unstructured pruning is well known that will not contribute to any acceleration in practice.
2. For structured pruning, the speedup ratio should use latency, not theoritical FLOPs. And more recent methods should be compared.
3. The visual prompting method essentially uses lower resolution for the input images. It is necessary to compare a baseline that using a lower resolution image as input, and then prune less parameters to maintain the same FLOPs as the proposed VPN.
4. Following the previous point, I also wonder whether this method will deteriorate some applications that are senstive to resolution, such as object detection or etc.

**Questions:**

See above.

---

> ### Author Response · Authors · 2023-11-18
> **Point-to-point Response to Reviewer eRqj (Part 1/3)**
>
> Thank you for recognizing that our idea is “easy to follow and effective” and our performance is “interesting”. To address reviewer eRqj’s concerns, we detailed our responses below.
>
> **[Cons 1. Focusing more on the structured sparse case.]**
>
> Thank you for the suggestion. We kindly point out that our main contributions focus on introducing the new data-model co-design pruning paradigm and we respectfully argue that focusing on unstructured pruning is highly meaningful. Unstructured pruning is extremely useful as both a mathematical prototype and an empirical testbed for new SNN algorithms and it is also receiving increasingly better support in practice. The evidence is as follows:
>
> 1. [Better performance than structured pruning] As the finest-grained and most flexible sparsity level, unstructured sparsity is superior to other more structured forms of sparsity [7].
>
> 2. [Widely used on nonGPU hardware] Unstructured sparsity has widely proven its practical relevance on nonGPU hardware, such as CPUs or customized accelerators. For instance, in the range of 70-90% high unstructured sparsity, XNNPACK [8] has already shown significant speedups over dense baselines on smartphone processors.
>
> 3. [Receive increasing support] The hardware support of unstructured sparsity may be relatively limited on “off-the-shelf” commodity GPUs/TPUs, but it keeps improving quickly over the years. For example, advanced GPU kernels such as NVIDIA cuSPARSE [9] and Sputnik [10] have built the momentum to better support finer-grained sparsity.
>
> 4. [Our method achieves time reduction in mask finding] We demonstrate that our method (unstructured pruning) achieves significant time reduction in terms of mask finding compared to other pruning methods like HYDRA and BiP as shown in [Figure 9](https://imgur.com/t2ysbTc).
>
> **[Cons 2. The speedup ratio should use latency.]**
>
> Thank you for the advice. We use FLOPs following [11, 12] and the FLOPs are also widely used in calculating the speedup ratio. To alleviate reviewer eRqj’s concern, we provide the latency speedup ratio of our method on the Quadro RTX 6000 GPU in Table R6. **Our method achieves latency speedup ratios of 1.1× and 1.2× at 10% and 20% channel-wise sparsity** respectively without compromising the performance relative to the dense network, as indicated in [Figure 8](https://imgur.com/VkoqArm). We kindly point out that our main contributions focus on introducing the new data-model co-design pruning paradigm and unstructured pruning, we provide the results of structured pruning only to indicate a potential research direction. We also include the latency results in the revised paper in Table A9.
>
> Table R6. The latency and FLOPs of VPNs structured pruning on ImageNet pre-trained ResNet-18 and CIFAR100.
> | Sparsity | Latency(ms) | FLOPs(G) |
> | :----------: | :----------: | :----------: |
> | Dense | 2.23 &plusmn; 0.03 | 1.82 |
> | 10% sparsity | 2.02 &plusmn; 0.02 | 1.68 |
> | 20% sparsity | 1.86 &plusmn; 0.02 | 1.45 |

---

> > ### Comment · Reviewer_eRqj · 2023-11-22
> >
> > 1. I would like to point out that even the XNNPACK mentioned that it needs block sparsity or certain 1x1 conv block as refered in [link](https://www.tensorflow.org/model_optimization/api_docs/python/tfmot/sparsity/keras/PruneForLatencyOnXNNPack) and [link](https://www.tensorflow.org/model_optimization/guide/pruning/pruning_with_sparsity_2_by_4). For NVidia GPU, it needs the 2 by 4 sparsity pattern mentioned in previous link. The rebuttal on structured pruning is an obvious sophism to me. The provided latency test is also only for channel pruning, which is a kind of structured pruning.

---

> ### Author Response · Authors · 2023-11-18
> **Point-to-point Response to Reviewer eRqj (Part 2/3)**
>
> **[Cons 3. More recent structured pruning methods should be compared.]**
>
> Thank you for the suggestion. We gently indicate that we have considered several recent state-of-the-art methods such as DepGraph which is a very strong baseline published in 2023 in Section 4.2, Paragraph 1, Line 3.
>
> **Additional experiments on structured pruning.** To further address reviewer eRqj’s concern, we conducted additional experiments on ImageNet pre-trained ResNet-18 and CIFAR100 using GReg [13] and LAMP [14] which are two recent structured pruning methods. We observe that **our method has the best performance** as shown in Table R7. We add the new results in our revised paper in Table A10. Due to the time limitation, we provide outcomes only at two sparsity levels, more sparsity levels are promised in our final version. If the reviewer can kindly point out more methods, we will also add them in our final version.
>
> Table R7. Performance comparison of our method, GReg, and LAMP on ImageNet pre-trained ResNet-18 and CIFAR100 at 20% and 50% channel-wise sparsity levels.
> | Method | 20% sparsity | 50% sparsity |
> | :----------: | :----------: | :----------: |
> | GReg | 78.91 | 62.63 |
> | LAMP | 80.7 | 75.46 |
> | VPNs (ours) | 81.21 | 75.58 |
>
> **[Cons 4. Our method essentially uses lower resolution input and needs to compare the baselines with lower resolution.]**
>
> This might be a misunderstanding of our setting and we respond to the concern from the following aspects:
>
> 1. [The resolution of our method and baselines is always 224] In our design, the resolution of our method is always 224 which equals the resolution of the input images in all baselines.  Although the visual prompt as learnable parameters are added to the margin of the image, the full image pixel information is preserved. Evidence can be found in Section 4.1, Paragraph 3, Line 5 that we use an input size of 224 and a pad size of 16, and in [Figure 3](https://imgur.com/y1fRg7v)  that i = 224 and p = 16.
>
> 2. [**Additional experiments of baselines using lower resolution and pruning fewer parameters**] To further alleviate reviewer eRqj's concern, we conducted additional experiments on some of the best baselines such as HYDRA and OMP with lower resolution and pruning fewer parameters on ImageNet pre-trained ResNet-18 and CIFAR100. HYDRA and OMP use an input resolution of 192 and prune 13k (the number of parameters in the visual prompt) fewer parameters than our method. We find that **our method achieves {3.94%, 3.69%} higher accuracy** than {HYDRA, OMP} at 90% sparsity level from the results displayed in Table R8, which indicates our method is superior to baselines using lower resolution and pruning fewer parameters. We also add the results in the revision of our paper in Table A11. Due to the time limitation, we provide outcomes only at two sparsity levels, more sparsity levels are promised in our final version.
>
> Table R8. Performance comparison of our method, HYDRA, and OMP on ImageNet pre-trained ResNet-18 and CIFAR100. HYDRA and OMP use 192 as the input resolution and prune 13k fewer parameters than our method.
> | Method | 50% sparsity | 90% sparsity |
> | :----------: | :----------: | :----------: |
> | HYDRA | 80.9 | 76.64 |
> | OMP | 80.89 | 76.89 |
> | VPNs (ours) | 83.18 | 80.58 |

---

> > ### Comment · Reviewer_eRqj · 2023-11-22
> >
> > I am afraid the authors misunderstand my questions about resolution. What I mean is that the effective content of image is of size 192x192. Thus we need to compare the results of VPN of size 224 (actual size 192) of sparsity x and other methods of size 192 of sparsity y, in which x and y are chosen that the compared methods have similar FLOPs, because the additional content in VPN is not necessary for other methods.

---

> ### Author Response · Authors · 2023-11-18
> **Point-to-point Response to Reviewer eRqj (Part 3/3)**
>
> **[Cons 5. Whether our method will deteriorate in applications that are sensitive to resolution such as object detection.]**
>
> This is an interesting direction to explore. We provide detailed responses below.
>
> 1. [Our method is insensitive to resolution] As illustrated above, our method and the baselines all use a resolution of 224 in the original setting and our method surpasses all baselines as shown in [Figure 4](https://imgur.com/YF7ODNx). In the additional experiments displayed in Table R8, our method is still better than the baselines using a resolution of 192. No matter when the baselines use the resolution of 224 or 192, our method is consistently better than the baselines, demonstrating our method is insensitive to resolution.
>
> 2. [**Superior performance on object detection**] To further alleviate reviewer eRqj's concern, we conducted supplemental experiments on Pascal VOC 2007, which is a well-known object detection task. We compare our method to HYDRA and OMP on YOLOv4 with ImageNet pre-trained ResNet-18 backbone. **Our method achieves {3.78%, 2.67%} higher AP** than {HYDRA, OMP} at 90% sparsity level as shown in Table R9, which indicates the superiority of our method in resolution-sensitive tasks. The results are also included in the revision of our paper in Table A6. Due to the time limitation, we provide outcomes only at two sparsity levels, more sparsity levels are promised in our final version.
>
> Table R9. AP comparison of our method, HYDRA, and OMP on YOLOv4 with ImageNet pre-trained ResNet-18 backbone and Pascal VOC 2007.
> | Method | 50% sparsity | 90% sparsity |
> | :----------: | :----------: | :----------: |
> | HYDRA | 35.25 | 32.74 |
> | OMP | 35.01 | 33.85 |
> | VPNs (ours) | 38.37 | 36.52 |
>
> [7] Mao, Huizi, Han, Song, Pool, Jeff, Li, Wenshuo, Liu, Xingyu, Wang, Yu & Dally, William J 2017 Exploring the granularity of sparsity in convolutional neural networks. In Proceedings of the IEEE Conference on Computer Vision and Pattern Recognition Workshops, pp. 13–20.
>
> [8] Elsen, Erich, Dukhan, Marat, Gale, Trevor & Simonyan, Karen 2020 Fast sparse convnets. In Proceedings of the IEEE/CVF conference on computer vision and pattern recognition, pp. 14629–14638.
>
> [9] Valero-Lara, Pedro, Martínez-Pérez, Ivan, Sirvent, Raül, Martorell, Xavier & Pena, Antonio J 2018 Nvidia gpus scalability to solve multiple (batch) tridiagonal systems implementation of cuthomasbatch. In Parallel Processing and Applied Mathematics: 12th International Conference, PPAM 2017, Lublin, Poland, September 10- 13, 2017, Revised Selected Papers, Part I, pp. 243–253. Springer.
>
> [10] Gale, Trevor, Zaharia, Matei, Young, Cliff & Elsen, Erich 2020 Sparse gpu kernels for deep learning. In SC20: International Conference for High Performance Computing, Networking, Storage and Analysis, pp. 1–14. IEEE.
>
> [11] Zhuang Liu, Jianguo Li, Zhiqiang Shen, Gao Huang, Shoumeng Yan, and Changshui Zhang. Learning efficient convolutional networks through network slimming. In Proceedings of the IEEE international conference on computer vision, pp. 2736–2744, 2017.
>
> [12] Gongfan Fang, Xinyin Ma, Mingli Song, Michael Bi Mi, and Xinchao Wang. Depgraph: Towards any structural pruning. In Proceedings of the IEEE/CVF Conference on Computer Vision and Pattern Recognition, pp. 16091–16101, 2023.
>
> [13] Huan Wang, Can Qin, Yulun Zhang, and Yun Fu. Neural pruning via growing regularization. arXiv preprint arXiv:2012.09243, 2020.
>
> [14] Jaeho Lee, Sejun Park, Sangwoo Mo, Sungsoo Ahn, and Jinwoo Shin. Layer-adaptive sparsity for the magnitude-based pruning. arXiv preprint arXiv:2010.07611, 2020.

---

> > ### Comment · Reviewer_eRqj · 2023-11-22
> >
> > How can you transfer your method to VOC, since the input size of detection tasks are usually much larger than that of classification. Moreover, VOC2007 IS NOT a valid dataset for evaluating detection tasks nowadays, larger dataset such as COCO is a must.

---

> ### Author Response · Authors · 2023-11-20
> **Response to Reviewer eRqj**
>
> Dear Reviewer **eRqj**,
>
> We thank reviewer **eRqj** time for the review and constructive comments. We really hope to have a further discussion with the reviewer **eRqj** to see if our response solves the concerns.
>
> In our response, we have (1) interpreted the contribution of our method and the significance of unstructured pruning; (2) conducted additional experiments on structured pruning, object detection, and baselines with lower resolution, further indicating the superiority of our method.
>
> We genuinely hope reviewer **eRqj** could kindly check our response. Thanks!
>
> Best wishes,
>
> Authors

---

> ### Author Response · Authors · 2023-11-22
> **Last Day Reminder**
>
> Dear Reviewer **eRqj**,
>
> We genuinely appreciate your dedicated time and effort in reviewing our work. As the final day of the discussion period approaches, we kindly request that you share any additional questions or concerns you may have. We are eager to engage in further discussions with you.
>
> If our responses have adequately addressed your concerns, we kindly ask that you consider raising the score of our work. Again, we are truly grateful for your valuable time and efforts.
>
> Warm regards,
>
> The Authors

---

> ### Comment · Reviewer_eRqj · 2023-11-22
> **After rebuttal score**
>
> Above all, the rebuttal has not addressed my concerns at all. I have to lower my score.

---

> > ### Author Response · Authors · 2023-11-23
> > **Point-to-point Response to Reviewer eRqj (Part 1/2)**
> >
> > **[Cons1. The rebuttal on structured pruning is an obvious sophism.]**
> >
> > We firmly disagree!
> >
> > 1. Reviewer eRqj seems to discourage the unstructured pruning research, which is highly biased and unfair.
> >
> > 2. As we explicitly pointed out in our rebuttal and main paper, our paper mainly focuses on unstructured pruning and the channel-wise structural pruning is our side study.
> >
> > 3. Unstructured pruning is meaningful! It achieves [**Better performance than structured pruning**] as indicated in [7]. The meaning of unstructured pruning can also be proved by that unstructured pruning serves as a technology basis for sparse learning of deep neural networks (DNNs). Besides efficiency, [**many other metrics are strongly related to sparsity**], *e.g.*, adversarial robustness [16], out-of-distribution generalization [17], and model transferability [18].
> >
> > 4. Unstructured pruning is [**Widely used on nonGPU hardware**]. A more notable success was recently demonstrated by DeepSparse [19, 20] which successfully deploys large-scale BERT-level sparse models on modern Intel CPUs, obtaining 10× model size compression with < 1% accuracy drop, 10× CPU-inference speedup with < 2% drop, and 29× CPU-inference speedup with < 7.5% drop.
> >
> > 5. Also, [**our method achieves time reduction in mask finding**] for our unstructured pruning methods. Although it is not our focus, we also report latency tests for structured pruning to answer eRqj's question.
> >
> > **[Cons2. About the resolution.]**
> >
> > Given the further clarifications of Reviewer eRqj’s resolution question. We address it below:
> >
> > 1. We have to restate that the effective resolution of our method and baselines is 224 but not 192 as indicated in Section 4.1, Paragraph 3, Line 5. Although the visual prompt as learnable parameters are added to the margin of the image, the full image pixel information is preserved.
> >
> > 2. We want to stress that our main contributions and results won’t be affected even if we conduct additional experiments on baselines with a resolution of 192 as reviewer eRqj asked.
> >
> >     - Our experiments are based on an entirely fair setting. As we stated in Section 4.1, Paragraph 3, Line 5, we unified the resolution of 224 in both our methods and the baselines. Therefore, the superiority of our proposed data-model co-design pruning paradigm indicated in Figure 4/5/6/7 in the revision is reliable.
> >
> >     - We believe that controlling the same FLOPs between our method using a resolution of 224 and the baselines using a resolution of 192 is NOT a conventional experiment setting, as we can hardly even find any references adopting this setting. Besides, our implementation follows the conventional pruning in [21, 22, 23], which compares different pruning methods under the same sparsity levels. Last but not least, it is not technically applicable to directly guarantee all the pruning methods under the same FLOPs budgets, since one can only calculate FLOPs until the subnetwork is determined. For pruning methods such as OMP and HYDRA, we need to finish the sparse training under many different sparsity levels and then choose the sparsity levels that have similar FLOPs.
> >
> > 3. Despite the difficulties, we conducted supplemental experiments, comparing our method using an input size of 224 and FLOPs of nearly 0.8G with baselines using an input size of 192 and FLOPs of nearly 0.8G. The experiments are carried out on ImageNer pre-trained ResNet-18 and CIFAR100 using our method, HYDRA, and OMP. We observe that our method outperforms HYDRA and OMP as shown in Table R17, which demonstrates our method is superior to baselines under the same FLOPs.
> >
> > Table R17. Performance comparison of our method, HYDRA, and OMP on ImageNet pre-trained ResNet-18 and CIFAR100. Our method uses 224 as the input resolution and HYDRA and OMP use 192 as the input resolution. Both our method and the baselines have FLOPs of 0.8G.
> > | Method | 0.8G FLOPs |
> > | :----------: | :----------: |
> > | HYDRA | 80.55 |
> > | OMP | 81.21 |
> > | VPNs (ours) | 82.40 |
> >
> > **[Cons3. How can we transfer our method to VOC since the input size of detection tasks is usually much larger than that of classification.]**
> >
> > We believe this is a misunderstanding of the design of VP (visual prompts). VP does NOT require a fixed input size of the pre-trained model, and therefore they can be used on both image classification [24] and object detection [25]. Therefore, although the input size of object detection is usually much larger than image classification, we can still transfer our method to VOC using a visual prompt design of a pad size of 32 and an input size of 416. In the additional experiments on object detection in Table R9, we follow the implementation in [26, 27] and the pruning process of our method is the same as that in image classification.

---

> > > ### Comment · Reviewer_eRqj · 2023-11-23
> > >
> > > The authors' emphasis on unstructured pruning in relation to BERT is intriguing. However, it's evident that visual prompting is not applicable to such a model. It also seems challenging for the authors to present a compelling example from the vision community in this context. Their defensiveness about this concept, which is well-acknowledged in the model acceleration community, is perplexing.
> > >
> > > Regarding the COCO experiments, I question the rationale behind highlighting a two-page technical report [27]. If the intent is to underscore that these seminal works also utilized the VOC dataset, it's noteworthy to mention that moco v1 [26] also reports COCO results alongside VOC. As a submission to a prestigious conference like ICLR, it's crucial to ensure the adequacy of experimental validation.
> > >
> > > In summary, while I recognize the potential of the ideas presented in this paper, it is imperative that the authors take the reviewers' comments seriously and make substantive improvements to their work.

---

> > ### Author Response · Authors · 2023-11-23
> > **Point-to-point Response to Reviewer eRqj (Part 2/2)**
> >
> > **[Cons4. Additional experiments on COCO.]**
> >
> > Given the limited time in the rebuttal period, we see the detection of VOC as a good and fair investigation to support the effectiveness of our proposal on applications that are sensitive to resolution [27].
> > We would like to point out that our paper's focus is to propose a superior pruning method from a data-model co-design perspective. We follow the standard experimental benchmark in this field [21, 22]. We would like to conduct additional experiments on COCO in our final version.
> >
> >
> >
> > [7] Mao, Huizi, Han, Song, Pool, Jeff, Li, Wenshuo, Liu, Xingyu, Wang, Yu & Dally, William J 2017 Exploring the granularity of sparsity in convolutional neural networks. In Proceedings of the IEEE Conference on Computer Vision and Pattern Recognition Workshops, pp. 13–20.
> >
> > [16] Sehwag, Vikash, et al. "Hydra: Pruning adversarially robust neural networks." Advances in Neural Information Processing Systems 33 (2020): 19655-19666.
> >
> > [17] Diffenderfer, James, et al. "A winning hand: Compressing deep networks can improve out-of-distribution robustness." Advances in Neural Information Processing Systems 34 (2021): 664-676.
> >
> > [18] Chen, Tianlong, et al. "The lottery tickets hypothesis for supervised and self-supervised pre-training in computer vision models." Proceedings of the IEEE/CVF Conference on Computer Vision and Pattern Recognition. 2021.
> >
> > [19] Kurtz, Mark, Kopinsky, Justin, Gelashvili, Rati, Matveev, Alexander, Carr, John, Goin, Michael, Leiserson, William, Moore, Sage, Nell, Bill, Shavit, Nir & Alistarh, Dan 2020 Inducing and exploiting activation sparsity for fast inference on deep neural networks. In Proceedings of the 37th International Conference on Machine Learning (ed. Hal Daumé III & Aarti Singh), Proceedings of Machine Learning Research, vol. 119, pp. 5533–5543. Virtual: PMLR.
> >
> > [20] Kurtic, Eldar, Campos, Daniel, Nguyen, Tuan, Frantar, Elias, Kurtz, Mark, Fineran, Benjamin, Goin, Michael & Alistarh, Dan 2022 The optimal bert surgeon: Scalable and accurate second-order pruning for large language models. arXiv preprint arXiv:2203.07259 .
> >
> > [21] Liu, Shiwei, et al. "The Unreasonable Effectiveness of Random Pruning: Return of the Most Naive Baseline for Sparse Training." International Conference on Learning Representations. 2021.
> >
> > [22] Tanaka, Hidenori, et al. "Pruning neural networks without any data by iteratively conserving synaptic flow." Advances in neural information processing systems 33 (2020): 6377-6389.
> >
> > [23] Sehwag, Vikash, et al. "Hydra: Pruning adversarially robust neural networks." Advances in Neural Information Processing Systems 33 (2020): 19655-19666.
> >
> > [24] Aochuan Chen, Yuguang Yao, Pin-Yu Chen, Yihua Zhang, and Sijia Liu. Understanding and improving visual prompting: A label-mapping perspective. In Proceedings of the IEEE/CVF Conference on Computer Vision and Pattern Recognition, pp. 19133–19143, 2023.
> >
> > [25] Liu, Weihuang, et al. "Explicit visual prompting for low-level structure segmentations." Proceedings of the IEEE/CVF Conference on Computer Vision and Pattern Recognition. 2023.
> >
> > [26] Kaiming He, Haoqi Fan, Yuxin Wu, Saining Xie, and Ross Girshick. Momentum contrast for unsupervised visual representation learning. In Proceedings of the IEEE/CVF Conference on Computer Vision and Pattern Recognition, pages 9729–9738, 2020.
> >
> > [27] Xinlei Chen, Haoqi Fan, Ross Girshick, and Kaiming He. Improved baselines with momentum contrastive learning. arXiv preprint arXiv:2003.04297, 2020.

---

### Official Review · Reviewer_sGzb · 2023-10-31

**Soundness:** 3 good
**Presentation:** 3 good
**Contribution:** 3 good
**Rating:** 5
**Confidence:** 4

**Summary:**

This paper proposes to use visual prompts to improve performance of the pruned model by applying visual prompts earlier in the process, aka, before the model fine-tuning. This effort was motivated by the experiments of applying post-pruning prompt to the sparse models with and without fine-tuning. As post-pruning prompts showed only marginal gains to subnets that went through fine-tuning, authors proposed to apply the visual prompts earlier in the process. The proposed scheme was compared with eight pruning baselines on eight classification tasks. Numerical comparisons show the potential of the proposed scheme.

**Strengths:**

- The idea of applying visual prompts to identify a subnet which further leads to a better pruning results is interesting.
- The idea of using visual prompts to control a pretrained vision model is an interesting direction to pursue.

**Weaknesses:**

- The paper is not easy to read. There's a particular emphasis on "data model co-design", but it takes quite a while to understand what this refers to concretely.
- Why is the visual prompts essential? How about learning additional parameters without using the visual prompt?
- How would visual prompts be different from data augmentation?

**Questions:**

Please see my questions in the weakness section.

---

> ### Author Response · Authors · 2023-11-18
> **Point-to-point Response to Reviewer sGzb (Part 1/2)**
>
> Thank you for recognizing that our results and the direction of our work are “interesting”. To address reviewer sGzb’s concerns, we provide detailed responses below.
>
> **[Cons 1. The statement of “data-model co-design” is not concrete.]**
>
> Thanks for putting forward the point. Specifically, data-model co-design indicates both the input data and the model are optimized in pruning. This paradigm of our method is significantly different from the baselines which are model-centric.
> 1. [Model-centric design] Focusing on searching and preserving crucial weights by analyzing network topologies (Abstract, Line 6) with fixed inputs as illustrated in [3].
> 2. [Data-model co-design] Optimizing weight masks and visual prompts jointly by combining data-centric design that constructs well-designed prompts (Section 1, Paragraph 3, Line 3) and model-centric design.
> More explanation of data-model co-design can be found in Section 1, Paragraph 5, Point 2; [Figure 2](https://imgur.com/tcdNFkO); and Section 3, Paragraph 1.
>
> To further alleviate reviewer sGzb’s concern, we significantly polished our paper in Section 1, Paragraph 5, and Section 3, Paragraph 1 by interpreting the data-model co-design more concretely to avoid misunderstanding in our revision.
>
> **[Cons 2. Why is the visual prompt essential?]**
>
> Thank you for the question. The significance of the visual prompt is detailed as follows:
>
> 1. [Motivation]
>
> - Xu et al. [4] demonstrate that prompts can recover compressed LLMs which illuminates the efficacy of post-pruning prompts in enhancing the performance of compressed LLMs. Observing that post-pruning prompts bolster the efficiency and performance of LLMs, It is only natural to inquire about the impact of VPs on vision models.
>
> - We observe from [Figure 1](https://imgur.com/37Np8Dh) that post-pruning prompts escalate the performance of the subnetworks before fine-tuning. However, neither of these settings consistently surpasses the standard no-prompting approach (pruning + fine-tuning). We postulate visual prompts are capable of enhancing the sparsification of vision models by utilizing them in a different way.
>
> 2. [Empirical evidence]
> - As shown in [Figure 4](https://imgur.com/YF7ODNx) and [Figure 5](https://imgur.com/7WL9Q9v), we demonstrate the essential role of VP by comparing our method (with VP) to HYDRA (without VP) on eight downstream datasets and three architectures.
>
> - As shown in [Figure 7](https://imgur.com/zqmXSrW), we observe that VP combined with existing prunings consistently surpasses their original counterpart (Section 4.2, Paragraph 4), which further illustrates the essential of VP.
>
> 3. [Previous literature]
>
> - [5, 6] showcases visual prompting as a parameter-efficient method that substantially improves the generalization and accuracy of vision models over multiple tasks. We believe these advantages brought by visual prompts also benefit pruning.

---

> ### Author Response · Authors · 2023-11-18
> **Point-to-point Response to Reviewer sGzb (Part 2/2)**
>
> **[Cons 3. How about learning additional parameters without using the visual prompt?]**
>
> Thank you for the suggestions. To the best of our understanding, the reviewer suggests comparing our method (with VP) to the baselines which learn more parameters than ours. Here is our response to the questions.
>
> 1. [Current results] As displayed in [Figure 4](https://imgur.com/YF7ODNx), our method achieves better performance at 80% sparsity than other baselines at {40%, 50%, 60%, 70%} sparsity levels on CIFAR100, CIFAR10, Food101, Flowers102, DTD, and StanfordCars.
>
> 2. [**Additional experiments of the baselines learning additional parameters**] We implemented supplemental experiments on ImageNet pre-trained ResNet-18 on CIFAR100, using our method, HYDRA, and OMP. HYDRA and OMP learn an additional 13k (the number of parameters in the visual prompt)  parameters than our method. We find that **our method obtains {3.46%, 3.06%} higher accuracy** than {HYDRA, OMP} at 90% sparsity level as displayed in Table R4, which manifests that our method is better than the baselines with additional parameters. The results are also reported in our revised paper in Table A7. Due to the time limitation, we provide outcomes only at two sparsity levels, more sparsity levels are promised in our final version.
>
> Table R4. Performance comparison of our method, HYDRA, and OMP on ImageNet pre-trained ResNet-18 and CIFAR100. HYDRA and OMP learn 13k additional parameters than our method.
> | Method | 50% sparsity | 90% sparsity |
> | :----------: | :----------: | :----------: |
> | HYDRA | 81.03 | 77.12 |
> | OMP | 81.36 | 77.52 |
> | VPNs (ours) | 83.18 | 80.58 |
>
> **[Cons 4. How would visual prompt be different from data augmentation?]**
>
> Visual prompt is essentially different from data augmentation, and the reasons are as follows:
>
> 1. [Purpose] Visual prompts are usually designed to improve model adaption. While data augmentations are used to avoid overfitting.
>
> 2. [Methodology] Visual prompts can be optimized in a data-driven way. However, most data augmentations are data-agnostic.In our method, visual prompting is the key to enabling the data-model exploration of crucial sparsity topologies.
>
> 3. [**Additional experiments of more data augmentations**] To further address reviewer sGzb’s concern, we compare our method without mix-ups to HYDRA and OMP with mix-ups on ImageNet pre-trained ResNet-18 and CIFAR10. **Our method achieves {5.46%, 3.50%} higher accuracy** than {HYDRA, OMP} at 90% sparsity level as shown in Table R5, which indicates that visual prompting is better than using more data augmentation. We add the results to the revised paper in Table A8. Due to the time limitation, we provide outcomes only at 50% and 90% sparsity levels, more sparsity levels are promised in our final version.
>
> Table R5. Performance comparison of our method (without mix-up), HYDRA (with mix-up), and OMP (with mix-up) on ImageNet pre-trained ResNet-18 and CIFAR10.
> | Method | 50% sparsity | 90% sparsity |
> | :----------: | :----------: | :----------: |
> | HYDRA (with mix-up) | 92.72 | 90.83 |
> | OMP (with mix-up) | 94.63 | 92.79 |
> | VPNs (without mix-up) | 96.47 | 96.29 |
>
>
> [3] Hidenori Tanaka, Daniel Kunin, Daniel L Yamins, and Surya Ganguli. Pruning neural networks without any data by iteratively conserving synaptic flow. Advances in neural information processing systems, 33:6377–6389, 2020.
>
> [4] Zhaozhuo Xu, Zirui Liu, Beidi Chen, Yuxin Tang, Jue Wang, Kaixiong Zhou, Xia Hu, and An-shumali Shrivastava. Compress, then prompt: Improving accuracy-efficiency trade-off of llm inference with transferable prompt. arXiv preprint arXiv:2305.11186, 2023.
>
> [5] Aochuan Chen, Yuguang Yao, Pin-Yu Chen, Yihua Zhang, and Sijia Liu. Understanding and improving visual prompting: A label-mapping perspective. In Proceedings of the IEEE/CVF Conference on Computer Vision and Pattern Recognition, pp. 19133–19143, 2023.
>
> [6] Kaiyang Zhou, Jingkang Yang, Chen Change Loy, and Ziwei Liu. Conditional prompt learning for vision-language models. In Proceedings of the IEEE/CVF Conference on Computer Vision and Pattern Recognition, pp. 16816–16825, 2022a.

---

> ### Author Response · Authors · 2023-11-20
> **Response to Reviewer sGzb**
>
> Dear Reviewer **sGzb**,
>
> We thank reviewer **sGzb** time for the review and constructive comments. We really hope to have a further discussion with the reviewer **sGzb** to see if our response solves the concerns.
>
> In our response, we have (1) interpreted the data-model co-design and the importance of visual prompts concretely; (2) conducted additional experiments on baselines learning more parameters and with more data augmentations, further indicating the superiority of our method.
>
> We genuinely hope reviewer **sGzb** could kindly check our response. Thanks!
>
> Best wishes,
>
> Authors

---

> ### Author Response · Authors · 2023-11-22
> **Last Day Reminder**
>
> Dear Reviewer **sGzb**,
>
> We genuinely appreciate your dedicated time and effort in reviewing our work. As the final day of the discussion period approaches, we kindly request that you share any additional questions or concerns you may have. We are eager to engage in further discussions with you.
>
> If our responses have adequately addressed your concerns, we kindly ask that you consider raising the score of our work. Again, we are truly grateful for your valuable time and efforts.
>
> Warm regards,
>
> The Authors

---

### Official Review · Reviewer_8b5d · 2023-11-01

**Soundness:** 3 good
**Presentation:** 4 excellent
**Contribution:** 3 good
**Rating:** 5
**Confidence:** 4

**Summary:**

This paper proposes a new post-pruning method by introducing visual prompts into the pruning pipeline. The authors introduce visual prompts into trained vision models. During pruning, not only weight masks but also visual prompts are optimized. The whole pipeline includes two steps: (1) fixing pretrained weights of vision models, tuning masks and visual prompts; (2) fixing the mask, fine-tuning both weights and visual prompts. The authors adopt the proposed method on several downstream classification tasks and compare the performances of pruned models with other pruning methods. The experiments show that the proposed method can achieve better performance than other methods with the same sparsity.

**Strengths:**

1. The writing and presentation of this paper are quite good. The logic of the article is very clear, and the choice of words in the writing is also very precise.
2. The idea of introducing tunable visual prompts into the pruning pipeline is intriguing. The experiments validate the effectiveness of the proposed strategy.
3. The authors compare the proposed method with other pruning methods and additionally apply the proposed visual prompt pruning strategy to these methods, demonstrating the transferability of their approach.

**Weaknesses:**

1. Introducing visual prompts into vision models seems to boost their performance. However, comparing models with visual prompts (the proposed method) to those without (other baseline methods) might not be entirely fair. What if we apply both the proposed and baseline methods to a model that has already been fine-tuned with visual prompts?
2. I have some doubts regarding the generality and performance of this paper.
(1) Why must we conduct experiments on downstream tasks of ImageNet? Why not directly on ImageNet itself, since most pruning work actually focuses more on performance on ImageNet?
(2) The method proposed in this paper seems to be limited to scenarios where visual prompts can be applied, with their main application currently being in classification tasks. How can the proposed approach be used for other tasks, such as detection, segmentation, etc.?

**Questions:**

Please refer to the weaknesses. I hope the authors can provide more experiments to demonstrate the effectiveness of the method.

The provided additional experiments have addressed my concerns about the performance of VPN in ImageNet and object detection tasks. However, after reading other reviewers' comments, I consider decreasing my score to 5. The reasons are as follows:
(1) The author describe the advantagement of unstructured sparsity on nonGPU hardware. However, they do not report the latency of unstructured sparsity on CPU. I think conducting experiments on some lightweight networks and reporting latency tested on CPUs can make the results more convincing.
(2) In regards to structured pruning, the experiments carried out on the CIFAR-100 dataset are insufficient.

Overall, I believe this paper needs more meticulous refinement in its experiments. If it can validate its approach on large-scale datasets for both unstructured and structured pruning settings, it would then be a very solid paper.

---

> ### Author Response · Authors · 2023-11-18
> **Point-to-point Response to Reviewer 8b5d**
>
> Many thanks to reviewer 8b5d for acknowledging that our proposal is “intriguing”, our experiments “validate the effectiveness” of our method, our writing and presentation are “quite good”, and we demonstrate the “transferability” of our approach. We sincerely appreciate all constructive suggestions, which help us to improve our paper further. To address reviewer 8b5d’s questions, we provide pointwise responses below.
>
> **[Cons 1. Comparing models with visual prompts to those without might not be entirely fair.]**
>
> We respectfully disagree. We have tried our best to guarantee fairness in the empirical results shown in [Figure 7](https://imgur.com/zqmXSrW). Specifically, in the setting of [Figure 7](https://imgur.com/zqmXSrW):
> 1. [For our method (VPNs)] We leverage VP in mask finding and subnetwork tuning.
> 2. [For baselines (VPNs w. Random, VPNs w. OMP, VPNs w. LTH)] We apply the same VP as our method to Random, OMP, and LTH in mask finding and subnetwork tuning.
>
> Both our method and baselines use exactly the same fine-tuning of subnetwork weights and VP. But our method apparently surpasses all baselines with visual prompting at all sparsity levels as shown in [Figure 7](https://imgur.com/zqmXSrW), which indicates the superiority of our method in the “entirely fair” setting.
>
> **Additional experiments of baselines with VP.** To further address reviewer 8b5d’s concern, we conduct supplemental experiments using our method and some of the best baselines such as LTH and OMP with VP on ImageNet pre-trained ResNet-18 network and Tiny-imageNet. We observe that **our method is superior to baselines with VP** as displayed in Table R1. We also include the results in the revision of our paper in Table A4. Due to the time limitation of rebuttal, we provide outcomes at 50% and 90% sparsity levels, and more sparsity levels are promised in our final version.
>
> Table R1. Performance comparison of our method, VPNs w. LTH, and VPNs w. OMP on ImageNet pre-trained ResNet-18 and Tiny-ImageNet.
> | Method | 50% sparsity | 90% sparsity |
> | :----------: | :----------: | :----------: |
> | VPNs w. LTH | 73.71 | 67.03 |
> | VPNs w. OMP | 73.56 | 64.78 |
> | VPNs (ours) | 73.82 | 67.89 |
>
> **[Cons 2. Why not directly conduct experiments on ImageNet itself?]**
>
> Thank you for the great advice. We provided results of pruning on ImageNet and fine-tuning on downstream datasets for transfer study in [Figure 6](https://imgur.com/pOVS2DX).
>
> **Additional experiments on ImageNet.** To further alleviate reviewer 8b5d’s concern, we conducted additional experiments on ImageNet using our method and some of the best baselines such as HYDRA and OMP, on ImageNet pre-trained ResNet-18 network. **Our method has {0.52%, 2.87%} higher accuracy** than {HYDRA, OMP} at 50% sparsity level as shown in Table R2, which illustrates the effectiveness of our method on ImageNet.  We also report the results in the revision of our paper in Table A5. Due to the time limitation of rebuttal, we provide outcomes at 50% and 90% sparsity levels, more sparsity levels are promised in our final version.
>
> Table R2. Performance comparison of our method, HYDRA, and OMP on ImageNet pre-trained ResNet-18 and ImageNet.
> | Method | 50% sparsity | 90% sparsity |
> | :----------: | :----------: | :----------: |
> | HYDRA | 68.91 | 66.62 |
> | OMP | 69.31 | 64.27 |
> | VPNs (ours) | 69.47 | 67.14 |
>
> **[Cons 3. How can the proposed approach be used for other tasks, such as detection?]**
>
> Thank you for the great point. Following the reviewer’s suggestion, we carried out additional experiments on Pascal VOC 2007 [1], which is one of the most widely used datasets for object detection tasks.
>
> **Additional experiments on Pascal VOC 2007.**  We compared our method to some of the best baselines such as HYDRA and OMP on YOLOv4 [2] which uses ImageNet pre-trained ResNet-18 as the backbone. The outcomes are presented in Table R3. **Our method achieves {3.78%, 2.67%} higher AP** than {HYDRA, OMP} at 90% sparsity level, which demonstrates the superiority of our method on object detection. We add the results to the revised paper in Table A6. Due to the time limitation, we provide outcomes only at two sparsity levels, more sparsity levels are promised in our final version.
>
> Table R3. AP comparison of our method, HYDRA, and OMP on YOLOv4 with ImageNet pre-trained ResNet-18 backbone and Pascal VOC 2007.
> | Method | 50% sparsity | 90% sparsity |
> | :----------: | :----------: | :----------: |
> | HYDRA | 35.25 | 32.74 |
> | OMP | 35.01 | 33.85 |
> | VPNs (ours) | 38.37 | 36.52 |
>
> [1] Mark Everingham, Luc Van Gool, Christopher KI Williams, John Winn, and Andrew Zisserman. The pascal visual object classes (voc) challenge. International journal of computer vision, 88(2):303–338, 2010.
>
> [2] Alexey Bochkovskiy, Chien-Yao Wang, and Hong-Yuan Mark Liao. Yolov4: Optimal speed and accuracy of object detection. arXiv preprint arXiv:2004.10934, 2020.

---

> > ### Comment · Reviewer_8b5d · 2023-11-22
> >
> > Thanks for you response. The additional experiments have addressed my concerns.

---

> > > ### Author Response · Authors · 2023-11-22
> > > **Thank you!**
> > >
> > > Thank you for kindly giving us feedback on our responses. We are glad to hear that your concerns have been addressed. We would like to thank you again for your help in improving our submission.

---

> ### Author Response · Authors · 2023-11-20
> **Response to Reviewer 8b5d**
>
> Dear Reviewer **8b5d**,
>
> We thank reviewer **8b5d** time for the review and constructive comments. We really hope to have a further discussion with the reviewer **8b5d** to see if our response solves the concerns.
>
> In our response, we have (1) clarified the fairness of our setting and carried out supplemental experiments on baselines with visual prompting; (2) conducted additional experiments on ImageNet and object detection tasks, further demonstrating the superiority of our method.
>
> We genuinely hope reviewer **8b5d** could kindly check our response. Thanks!
>
> Best wishes,
>
> Authors

---

> ### Author Response · Authors · 2023-11-22
> **Last Day Reminder**
>
> Dear Reviewer **8b5d**,
>
> We genuinely appreciate your dedicated time and effort in reviewing our work. As the final day of the discussion period approaches, we kindly request that you share any additional questions or concerns you may have. We are eager to engage in further discussions with you.
>
> If our responses have adequately addressed your concerns, we kindly ask that you consider raising the score of our work. Again, we are truly grateful for your valuable time and efforts.
>
> Warm regards,
>
> The Authors

---

### Author Response · Authors · 2023-11-18
**Highlighted General Response**

We sincerely appreciate all reviewers’ time and efforts in reviewing our paper. We also thank all reviewers for the insightful and constructive suggestions, which helped a lot in further improving our paper.

In addition to the pointwise responses below, here we summarize our updates.

1. **[Additional Experiments]**

As mentioned by the reviewers, we conduct new experiments to evaluate the superiority of our approaches on new tasks, datasets, architectures, and other circumstances. We list some of the experiments mentioned by multiple reviewers in the following.

- **ImageNet.** We conduct new experiments using our method, HYDRA, and OMP on ImageNet pre-trained ResNet-18 and fine-tuned on ImageNet, the results shown in Table R13 demonstrate **our method achieves superior performance on ImageNet**. We also report the results in the revision of our paper in Table A5. Due to the time limitation of rebuttal, we provide outcomes only at 50% and 90% sparsity levels, more sparsity levels are promised in our final version.

- **Object Detection.** We carry out additional experiments on Pascal VOC 2007, which is a widely used dataset for object detection tasks. We compare our method to some of the best baselines such as HYDRA and OMP on YOLOv4 which uses ImageNet pre-trained ResNet-18 as the backbone. The outcomes are presented in Table R14. **Our method achieves {3.78%, 2.67%} higher AP than {HYDRA, OMP}** at 90% sparsity level, which demonstrates the superiority of our method on object detection. We also add the results to the revised paper in Table A6. Due to the time limitation, we provide outcomes only at two sparsity levels, more sparsity levels are promised in our final version.

Table R13. Performance comparison of our method, HYDRA, and OMP on ImageNet pre-trained ResNet-18 and ImageNet.
| Method | 50% sparsity | 90% sparsity |
| :----------: | :----------: | :----------: |
| HYDRA | 68.91 | 66.62 |
| OMP | 69.31 | 64.27 |
| VPNs (ours) | 69.47 | 67.14 |

Table R14. AP comparison of our method, HYDRA, and OMP on YOLOv4 with ImageNet pre-trained ResNet-18 backbone and Pascal VOC 2007.
| Method | 50% sparsity | 90% sparsity |
| :----------: | :----------: | :----------: |
| HYDRA | 35.25 | 32.74 |
| OMP | 35.01 | 33.85 |
| VPNs (ours) | 38.37 | 36.52 |

2. **[Clearance]**

Questioned by the reviewers, we clear some important details in our experiments here.

**Our method and the baselines all use a resolution of 224.** In our design, the resolution of our method is always 224 which equals the resolution of the input images in all baselines.  Although the visual prompt as learnable parameters are added to the margin of the image, the full image pixel information is preserved. Evidence can be found in Section 4.1, Paragraph 3, Line 5 that we use an input size of 224 and a pad size of 16 and [Figure 3](https://imgur.com/y1fRg7v)  that i = 224 and p = 16.

3. **[Summary of Paper Revision]**

The revision of the paper is updated, including all new experiment results, clearances, new plots, and references. All changes are marked in blue. We will keep updating for better and clearer readability.

We hope our pointwise responses below can clarify all reviewers’ confusion and alleviate all concerns. We thank all reviewers’ time again.

---

### Meta-Review · Area_Chair_hzu5 · 2023-12-10

**Metareview:**

Prior work on neural network sparsification focused on the trained model - they formulated schemes to identify the most important parts of the network and removing the least important parts. This submission also explores the role of data in network sparsification. They propose the use of "visual prompts" which are learnable parameters on the periphery of the input image. The image mask and the neural network sparsification mask are jointly optimized. The authors show improvements in accuracy at a given network sparsification when using this learnable "visual prompt".
The idea of visual prompting based on learnable periphery values is not new and the authors, to their credit, have appropriately cited and referred to prior work in this area. However, the application to network pruning is new.
The reviewers raised issues with experiments conducted in the submission - they felt the range of experiments was insufficient and not convincing enough to support the claims.
After the AC-reviewer discussion, the most positive reviewer reduced their score from 6 to 5, resulting in a final distribution of 3 x 5 and 1 x 3. Multiple reviewers specifically had an issue with the latency measurements, and the authors are advised to better explain their testing methodology. The AC did not find sufficient reason to overturn the negative consensus of 4 reviewers.

**Justification For Why Not Higher Score:**

The submission initially received 1 x 6, 2 x 5, and 1 x 3. After discussion, one reviewer reduced their score from 6 to 5, leading to a final score of 3 x 5 and 1 x 3. The AC did not find sufficient reason to overturn the negative consensus.

**Justification For Why Not Lower Score:**

N/A

---

### Decision · Program_Chairs · 2024-01-16

Reject